# Unfolding Generative Flows with Koopman Operators: Trajectory-Preserving Linearization

**Erkan Turan** [1]  **Ari Siozopoulos** [1 2 3]  **Louis Martinez** [1]  **Julien Gaubil** [1]  **Emery Pierson** [1]  **Maks Ovsjanikov** [1]

## Abstract

Continuous Normalizing Flows (CNFs) enable elegant generative modeling but remain bottlenecked by their iterative nature requiring costly sampling and lacking interpretability of the intermediate states. Recent approaches accelerate sampling by straightening trajectories or distilling endpoints, yet they treat the original generative process as a black box, discarding the teacher's intermediate dynamics. We propose a fundamentally different perspective: globally linearizing flow dynamics via Koopman theory to achieve trajectory-preserving linearization. By lifting a pre-trained Conditional Flow Matching (CFM) model into a higher-dimensional Koopman space, we represent its evolution with a single linear operator. Crucially, unlike boundary-only distillation, our method enforces infinitesimal consistency with the teacher's vector field along *the full generative path*. We derive a practical, simulation-free training objective that ensures this global alignment and yields two key benefits. First, sampling becomes one-step and parallelizable. Second, because the linearization is faithful to the dynamics, the Koopman operator provides unique insights on the generation. We demonstrate that this structure enables novel applications unavailable in prior approaches, including discovery of semantically coherent editing directions, inversion with a teacher-aligned linear operator and class-conditional spectral signatures. Empirically, our approach achieves competitive sample quality, while enabling spectral analysis and control of the *entire trajectories* of generative flows.

## 1. Introduction

Modern generative modeling has been revolutionized by dynamical system-based approaches, such as Diffusion Models (Ho et al., 2020; Song et al., 2021) and Continuous Normalizing Flows (CNFs) (Chen et al., 2018), which achieve state-of-the-art sample fidelity. Yet, these models share a fundamental limitation: their dynamics are inherently non-linear.

This distinction matters because linear and non-linear systems occupy opposite ends of the analyzability spectrum. Linear dynamical systems are transparent and admit a powerful analytical toolkit: spectral analysis decomposes solutions into independent modes, matrix exponentials yield closed-form trajectories, and eigendecomposition reveals stability and long-term behavior at a glance. Furthermore, principles like superposition and variation of parameters apply universally.

Non-linear systems offer no such luxury. While *local* linearization is trivial, a Taylor expansion around any fixed point, this tells us *little or nothing about global behavior*. Trajectories can diverge, bifurcate, or exhibit chaos far from equilibrium. The rich phenomena of non-linear dynamics come at the cost of interpretability: we lack general tools to characterize the system as a whole.

Generative flows lie squarely in the non-linear regime. Techniques like Rectified Flow (Liu et al., 2023) and Optimal Transport CFM (Tong et al., 2024; Pooladian et al., 2023) use straighter paths, mainly to accelerate sampling, but the underlying prediction remains a non-linear black box. We can probe the model locally, but understanding *how* it transforms noise into data across the full trajectory remains elusive. Our main question stems as follows: Can we find a global linearization, a single fixed linear operator whose dynamics are consistent with the full non-linear generative path, not just local approximations around isolated points?

Such a bridge exists via Koopman Operator Theory, which lifts non-linear dynamics into a higher-dimensional space where evolution becomes linear (Koopman, 1931; Brunton et al., 2022). This theory, which has seen a resurgence with modern machine learning (Lusch et al., 2018; Otto & Rowley, 2019; Azencot et al., 2020), can be adapted to our

[1]LIX, Ecole Polytechnique, IP Paris [2]University of Athens, Greece [3]Archimedes/Athena RC, Greece. Correspondence to: Erkan Turan <turan@lix.polytechnique.fr>.

*Proceedings of the $43^{rd}$ International Conference on Machine Learning*, Seoul, South Korea. PMLR 306, 2026. Copyright 2026 by the author(s).

problem. Recently, (Berman et al., 2025) used this theory to accelerate diffusion sampling by learning a linear noise-to-data map, which ultimately behaves similarly as one-step samplers (Geng et al., 2025). There is no guarantee that intermediate dynamics align with the teacher's vector field.

Our contribution [1] is a framework that linearizes the *full trajectory*. We ground Koopman theory in the non-autonomous dynamics of Conditional Flow Matching (CFM) and derive a simulation-free training objective enforcing infinitesimal consistency along the entire generative path. Specifically:

1. We introduce a novel framework for learning a global Koopman linearization of the non-autonomous dynamics in pretrained Conditional Flow Matching models.

2. We derive a practical, simulation-free training objective that enforces consistency *along the full generative trajectory,* yielding a full linearization rather than mere boundary-focused distillation.

3. We demonstrate empirically that our method uniquely enables spectral analysis, disentangled generative control, and improved robustness in downstream tasks, while maintaining competitive CFM sampling performance.

## 2. Related Work

Our work connects two main areas: flow-based generative models, methods for accelerated sampling of iterative models, Koopman operator theory for dynamical systems, and interpretability in generative modeling.

**Flow-Based Generative Models.** Flow-based models learn an invertible mapping between a data distribution and a simple base distribution, offering tractable likelihoods (Dinh et al., 2015; 2017; Kingma & Dhariwal, 2018). Continuous Normalizing Flows (CNFs) parameterize this map as the solution to an ODE (Chen et al., 2018). Although powerful, training early CNFs was often unstable and computationally intensive. Conditional Flow Matching (CFM) represents a major step forward, providing a stable and efficient simulation-free training objective by regressing a neural network to a conditional vector field (Lipman et al., 2023; Tong et al., 2024; Liu et al., 2023). However, while these models have achieved high accuracy for generative modeling, their sampling process remains inherently slow. A popular direction to tackle this problem is to learn "straighter paths", as in Rectified Flow (Liu et al., 2023) and OT-CFM (Pooladian et al., 2023). However, those approaches only reduce the number of integration steps and ultimately remain a nonlinear black box.

[1]The code used for this paper can be found here: https://github.com/limesqueezy/unfolding-flows

**Distillation of flow models.** The slow and iterative sampling of CNFs has motivated extensive research to distill their knowledge into easy-to-use models. A popular approach consists of training a separate student model capable of single-step generation, including Consistency Models (Song et al., 2023; Kim et al., 2024) and other distillation techniques (Salimans & Ho, 2022; Luo et al., 2023; Liu et al., 2025). Recently, MeanFlow (Geng et al., 2025) proposed an elegant way to overcome the need for a teacher model by modeling the average velocity of the CNF. These methods achieve remarkable speed, their main objective. However, as straight-flow models, they typically yield a compressed, black-box sampler. Worse, they have no fidelity to the initial dynamics. In contrast, our Koopman framework provides both interpretability and analytical control on the original dynamic of the flow model.

**Interpreting and Explaining Generative Models.** While methods exist for interpreting the latent spaces of classic models, such as VAEs (Kim & Mnih, 2018; Khemakhem et al., 2020; Higgins et al., 2017; Burgess et al., 2017) and GANs (Chen et al., 2016; Jahanian et al., 2020; Härkönen et al., 2020; Voynov & Babenko, 2020; Shen & Zhou, 2021), applying such approaches to iterative generative models is not straightforward. Existing approaches for iterative generative models are often more complicated than the earlier methods (Kwon et al., 2023; Yang et al., 2023; Meng et al., 2022; Kulikov et al., 2025) (See Appendix G for an extended review). Another possibility is to analyze the operation regimes of generative models. (Biroli et al., 2024) decompose the generation process in time between class fidelity and image realism. (Bonnaire et al., 2025) show how the dataset construction affects the training of diffusion models. These analyses provide elegant theoretical insights under high-dimensional or simplified score assumptions.

We take a complementary, data-driven path rooted in rigorous Koopman Operator theory. Our global linearization yields a Koopman latent space *endowed with the structure of the underlying dynamics*, where classic linear editing is accessible. Semantic directions discovered in this space can be pulled back to pixel space, enabling controlled generation with the original CFM model.

## 3. Mathematical Background

### 3.1. Problem statement

A Continuous Normalizing Flow (CNF) maps a prior distribution $p_0$ to a data distribution $p_1$ by solving the ODE

$$\frac{dx_t}{dt} = v_t(x_t), \text{s.t. } x_0 \sim p_0, x_1 \sim p_1, \tag{1}$$

where $v_t$ is a time-dependent vector field (Chen et al., 2018). Conditional Flow Matching (CFM), proposes a tractable way of learning a CNF (see Appendix A.1 for more details).

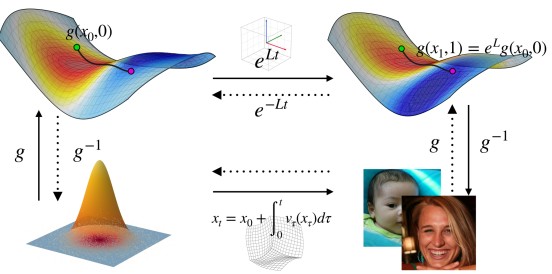

*Figure 1.* Overview of our method: We propose to learn a Koopman latent space of "observables", where the nonlinear dynamics of a CFM teacher becomes linear (Section 4.). Any point in the trajectory is now accessible with a single matrix exponential step of the Koopman operator L, including noise-to-image generation and image-to-noise inversion. The operator L can be decomposed and used to analyze the CFM teacher (Section 5.)

Sampling from a trained CFM model requires numerically integrating its ODE via $x_1 = x_0 + \int_0^1 v_\theta(s, x_s)ds$, a slow process with potentially many function evaluations (Chen et al., 2018).

A common solution is to learn a direct mapping $x_1 = f_\theta(x_0)$. One approach learns $f_\theta$ by regressing on endpoint pairs $(x_0, x_1)$ (Berman et al., 2025); another enforces straight paths constraints (Liu et al., 2023; Geng et al., 2025). Neither addresses our full goal. The first learns only the integrated map $x_1 = x_0 + \int_0^1 v_\tau(x_\tau)\,d\tau$, which is agnostic to the velocity field itself since any $v_t' = v_t + \frac{df}{dt}$ with $f(0) = f(1) = 0$ yields the same endpoints. The second replaces the original dynamics with straight trajectories rather than analyzing them. We seek a framework that linearizes *any* given dynamics while remaining faithful to its vector field throughout generation.

Our objective is to find a finite set of invertible observables $y_t = g_t^\theta(x_t) \in \mathbb{R}^m$, such that *the full dynamics are linear*, i.e, there exists a matrix $A$ such that :

$$\frac{dy_t}{dt} = Ay_t, \text{s.t. } y_0 = g_0(x_0), x_0 \sim p_0 \quad (2)$$

Crucially, we require $A$ to be *consistent* with the original velocity field $v_t$: the linear evolution in observable space must correspond to the non-linear flow in data space at *all times*, not merely at endpoints. Such a formulation yields two key advantages: first, sampling becomes analytical: $x_t = g^{-1}(y_t)$, where $y_t = e^{tA}y_0$, and the dynamics become decomposable into different modes by using an adapted eigendecomposition of $A$. We do so by building on top of Koopman theory, whose central idea is to shift perspective from the finite-dimensional state space to the infinite-dimensional space of functions - the "observables" - where the dynamics become linear.

## 3.2. Koopman Theory for autonomous systems

Koopman theory provides a powerful framework for globally linearizing nonlinear dynamical systems (Koopman, 1931; Mezić, 2005). For an overview of the field, we refer the interested reader to the excellent introduction by (Brunton et al., 2022).

Formally, consider an autonomous dynamical system $\frac{dx_t}{dt} = v(x_t)$. This system induces a *flow map* $F_t$ that advances an initial state $x$ to its value at time $t$, namely $x_t = F_t(x)$, along the trajectories defined by $v$. Let $g : \mathbb{R}^d \to \mathbb{R}$ be an *observable function* on the state space. Given an initial state $x$, the *Koopman operator* $\mathcal{K}_t$ is a mapping on the space of observables, denoted $\mathcal{G}(\mathbb{R}^d)$, which evolves observables along the trajectories generated by the vector field $v$:

$$\mathcal{K}_t g(x) := (g \circ F_t)(x) = g(F_t(x)) = g(x_t). \quad (3)$$

Note that $\mathcal{K}_t$ takes as input a real-valued function $g$ and produces another real-valued function. Koopman theory builds on the fact that this operator is trivially linear (regardless of the non-linearity of $F_t$) due to the linearity of the composition of functions: $K_t(g_1 + g_2)(x) = (g_1 + g_2) \circ F_t(x) = g_1 \circ F_t(x) + g_2 \circ F_t(x) = \mathcal{K}_t g_1(x) + \mathcal{K}_t g_2(x)$, for all observables $g_1, g_2$.

Taking the Lie derivative, we can then define the **Koopman generator**, $\mathcal{L}$, such that $\mathcal{L}g := \lim_{t \to 0} \frac{\mathcal{K}_t g - g}{t}$, and one can show that (Brunton et al., 2022)

$$\mathcal{L}g = \frac{dg}{dt} = \nabla_x g(x) \cdot v(x), \quad (4)$$

which is also trivially linear in $g$, leading to a linear equation on the space of observables. The operator and generator are related by the operator exponential, $\mathcal{K}_t = \exp(t\mathcal{L})$. Finding $\mathcal{L}$ is the objective of Koopman theory.

In summary, the potentially complex and non-linear ODE Equation (1) on the finite-dimensional state space $\mathbb{R}^d$ can be expressed as a linear equation in another space, $\mathcal{G}(\mathbb{R}^d)$, which consists of scalar-valued functions defined on the state space. The practical challenge in Koopman theory is to find *invertible mappings* $f : \mathbb{R}^d \to \mathcal{G}(\mathbb{R}^d)$ that allow solving the linear equation in the observable space and then recovering the solution in the original state space. However, computing such a mapping is often intractable in practice due to the *infinite dimensionality* of $\mathcal{G}(\mathbb{R}^d)$.

A particular case arises when there exists an $m$-dimensional linear subspace of $\mathcal{G}(\mathbb{R}^d)$, $F = \text{span}\{g_i\}_{i=1}^m$, invariant under the linear operator $\mathcal{L}$. The action of the generator on $F$ can then be represented by a single finite-dimensional matrix $L \in \mathbb{R}^{m \times m}$. The dynamics on this space of observables can then be written as:

$$\frac{d\mathbf{g}_t}{dt}(x) = L\mathbf{g}_t(x), \quad (5)$$

where $\mathbf{g}_t(x) = [g_1(x_t), \ldots, g_m(x_t)]^\mathsf{T} \in \mathbb{R}^m$ are the *Koopman coordinates*, i.e., the values of the observables $\{g_i\}_{i=1}^m$ evaluated at the state $x_t$, where $x_t$ is the evolution of the initial state $x$ to time $t$ along the trajectories generated by the dynamics.

Thus, the general goal when applying Koopman theory to dynamical systems is to (1) identify a sufficiently expressive set of observables $\{g_i\}_{i=1}^m$ and (2) determine the Koopman generator matrix $L$ on this space of observables.

In light of this theory, we ask:
1. For generative flows, does such a family of observable functions $\{g_i\}_{i=1}^m$ and corresponding generate exist?
2. Can we learn them in an efficient and unbiased way?

We expose our methodology and theoretical results in the next Section to answer these questions.

# 4. Methodology and Theoretical Results

Our objective is to learn a Koopman representation for a pre-trained CFM model, specified by its vector field $v_t$. This involves learning an encoder $g_\phi$ as the Koopman observables that linearizes the dynamics, a generator matrix $L$, and a decoder $g_\psi^{-1}$ that maps back to the state space. Here $\phi$ and $\psi$ are the learnable parameters of the corresponding neural networks. Several challenges arise compared to previous neural Koopman-based approaches (Lusch et al., 2018):

1. CFM dynamics are non-autonomous (explicitly time-dependent), whereas classic Koopman theory applies to autonomous systems (Section 4.1).

2. The learned observables $g$ must be expressive enough to capture the dynamics and reproduce trajectories and sampling accurately (Section 4.2).

3. The training objective for the Koopman representation must be tractable, ideally inheriting the simulation-free nature of CFM (Section 4.3).

## 4.1. Adapting Koopman Theory to Non-Autonomous Dynamics

**Time-dependent transformation.** As mentioned above, Koopman theory applies to autonomous dynamics, where the velocity $v(x_t)$ does not depend on the time. We can address this time-dependence of $v_t(x_t)$ by using a standard transformation used in the dynamical systems literature ((Strogatz, 2024), Chap 1.): we augment the state space to include time. The state becomes $y_t = (t, x_t)$, and the dynamics are defined on this augmented space with respect to a new external time parameter $\tau$:

$$\frac{dy}{d\tau} = \frac{d(t, x_t)}{d\tau} = [1, \ v_t(x_t)]. \tag{6}$$

Our observables are now functions of both space and time, $g(t, x)$. A crucial detail, however, is how we parameterize the linear dynamics on this augmented state to ensure the time variable evolves correctly (i.e., $\dot{t} = 1$).

**Affine lift for time evolution.** To enforce the constraint $\dot{t} \equiv 1$, we use an *affine lift*. The state is augmented with a constant bias coordinate to become $\mathbf{z}_t = [1, \ t, \ g(t, x)]^T$. For the dynamics $\dot{\mathbf{z}} = L\mathbf{z}$ to satisfy the physical constraints $\dot{1} = 0$ and $\dot{t} = 1$ for all states, the generator $L$ is uniquely constrained to adopt a block structure:

$$\tilde{L} = \begin{bmatrix} 0 & 0 & \mathbf{0} \\ 1 & 0 & \mathbf{0} \\ \mathbf{b}_g & \mathbf{A}_{gt} & \mathbf{A}_{gg} \end{bmatrix} \tag{7}$$

This parameterization guarantees correct time evolution by design and yields affine dynamics for the observables: $\dot{g} = \mathbf{b}_g + \mathbf{A}_{gt}t + \mathbf{A}_{gg}g$. The learned parameters are the weights $\phi, \psi$ of the encoder $g_\phi$ and decoder $g_\psi^{-1}$ and the matrix blocks $(\mathbf{b}_g, \mathbf{A}_{gt}, \mathbf{A}_{gg})$.

## 4.2. Learning Koopman Dynamics

Given a pre-trained CFM teacher network $v_t$, our main goal is to learn observable functions $\{g_i\}_{i=1}^m$ that span a finite-dimensional subspace *invariant under* the Koopman generator $\mathcal{L}$ *associated with* the dynamics $v_t$, and to learn the corresponding generator on this space. We learn the observables with an encoder $g_\phi$ that maps an initial state $x \in \mathbb{R}^d$ to its Koopman coordinates at time $t$, $\mathbf{g}_t(t, x) = [g_1(t, x_t), \ldots, g_m(t, x_t)]^\mathsf{T} \in \mathbb{R}^m$. We also learn the Koopman generator on this space as a dense matrix $L \in \mathbb{R}^{m \times m}$. To recover the solution of the ODE in the original state space and ensure the learned linear dynamics correspond to the *underlying nonlinear dynamics*, we also learn a decoder network $g_\psi^{-1}$ that maps the Koopman coordinates $\mathbf{g}_t(x)$ back to the state $x_t$ at time $t$.

We generate noise and target-data pairs $(x_0, \ x_1)$ using the pretrained CFM model, and aim to learn the following mapping:

$$x_t \simeq g^{-1}(e^{tL}g(0, \ x_0)).$$

**Training loss** Our training objective is as follows:

$$\mathcal{L}_{\text{train}} = \lambda_{\text{phase}}\mathcal{L}_{\text{phase}} + \lambda_{\text{target}}\mathcal{L}_{\text{target}} + \lambda_{\text{recon}}\mathcal{L}_{\text{recon}} + \lambda_{\text{cons}}\mathcal{L}_{\text{cons}}.$$

The first two terms ensure that the integrated linear dynamics map the start of a trajectory to its end in the Koopman space (phase loss):

$$\mathcal{L}_{\text{phase}} = \mathbb{E}_{(x_0, \ x_1)} \left\| e^L g_\phi(0, \ x_0) - g_\phi(1, \ x_1) \right\|^2, \tag{8}$$

and in the state space (after decoding - target loss):

$$\mathcal{L}_{\text{target}} = \mathbb{E}_{(x_0, \ x_1)} \left\| g_\psi^{-1}\left(e^L g_\phi(0, \ x_0)\right) - x_1 \right\|^2, \tag{9}$$

The third term encourages that we can retrieve the final state with the decoder:

$$\mathcal{L}_{\text{recon}} = \mathbb{E}_{x_1} \left[ d_{\text{Image}} \left( g_\psi^{-1} \left( g_\phi(1, \ x_1) \right), x_1 \right) \right] \quad (10)$$

where $d_{\text{Image}}$ is a distance measure on the image space, such as MSE or LPIPS (Zhang et al., 2018). The reconstruction loss is particularly important due to an inherent non-identifiability in the Koopman representation, as formalized in the proposition below. This term allows us to find, among the space of Koopman linearizing coordinate systems, the decodable ones.

We limit the reconstruction penalty to $t = 1$ to focus the decoder's capacity on clean images. We observed that this constraint is sufficient to identify the Koopman coordinates, leaving the model free to enforce linear dynamics across the rest of the trajectory (which is promoted using $\mathcal{L}_{\text{cons}}$ described below) without focusing on potentially restrictive intermediate noise decoding.

**Proposition 1** (Non-identifiability up to linear transformation). *The Koopman observable coordinates $g$ are identifiable only up to an arbitrary invertible linear transformation $M$. If the pair $(g, L)$ satisfies the consistency and phase objectives, so does the transformed pair $(M^{-1}g, \ M^{-1}LM)$.*

**Corollary 1.1.** A reconstruction loss of the form $\|g^{-1}(g(t, \ x)) - x\|^2$, with a decoder $g^{-1}$, breaks this invariance. It "fixes the gauge" by selecting the specific coordinate system that the chosen decoder can successfully map back to the data space.

The proof is provided in Appendix A. This result motivates the necessity of $\mathcal{L}_{\text{recon}}$ to obtain a unique and useful representation.

### 4.3. Efficient Dynamics learning

The combination of the three losses described above is necessary to find suitable observables, and matches the approach of (Berman et al., 2025) closely, except that our formulation is continuous and theirs is discrete. The main difference comes from our *trajectory consistency loss*, which forces the dynamics in the learned latent space to be governed by the linear generator $L$, by adapting Equation (4) to our problem:

$$\mathcal{L}_{\text{cons}} = \mathbb{E}_{t, \ x_t \sim p_t(x_t)} \|Lg_\phi(t, \ x_t) - \nabla_x g_\phi(x_t) \cdot v_t(x_t)\|^2. \quad (11)$$

One might notice that, similarly to the CNF setting, the consistency loss $\mathcal{L}_{\text{cons}}$ written in Eq. equation 11 is potentially intractable, as it would require sampling from the path distribution $x_t \sim p_t(x_t)$. A first solution would be to generate full trajectories $(x_t)_t$, but this would pose both discretization and scale problems for storing the pre-computed trajectories. Another solution is to hope to substitute the marginal

velocity $v_t(x_t)$ with the conditional velocity $u_t(x_t|x_1)$ and sample from the tractable $p_t(x_t|x_1)$, mirroring the CFM training strategy. However, as the following proposition shows, these two objectives are not equivalent when learning the encoder $g$.

**Proposition 2** (Marginal vs. Conditional Objectives). *Let $\mathcal{L}_{marg}$ be the desired consistency loss evaluated over the marginal distribution $p_t(x_t)$, and let $\mathcal{L}_{cond}$ be the tractable alternative evaluated using conditional samples and velocities. The two objectives are related by:*

$$\mathcal{L}_{cond} = \mathcal{L}_{marg} + \Delta(g), \quad (12)$$

*where,*

$$\Delta(g) = \mathbb{E}_{t, x_1, x_t} \left\| \nabla_x g(t, x_t) \left( u_t(x_t \,|\, x_1) - v_t(x_t) \right) \right\|^2 \ \geq \ 0.$$

The proof is provided in Appendix A. Because of the positive, $g$-dependent term $\Delta(g)$, minimizing $\mathcal{L}_{\text{cond}}$ will not necessarily minimize $\mathcal{L}_{\text{marg}}$.

Fortunately, as we have a pre-trained CFM model, the marginal velocity field $v_t(x_t)$ is known. This allows us to formulate a practical estimator for the true marginal loss, as stated in the following proposition.

**Proposition 3** (Practical Estimator for the Consistency Loss). *Given that the marginal path distribution $p_t(x_t)$ is defined as $p_t(x_t) = \int p_t(x_t|x_1)q(x_1)dx_1$, the marginal consistency loss $\mathcal{L}_{cons}$ can be estimated tractably using samples from the data distribution $q(x_1)$ and the conditional path $p_t(\cdot|x_1)$ as follows:*

$$\mathcal{L}_{cons} = \mathbb{E}_{t, x_1 \sim q, x_t \sim p_t(\cdot|x_1)} \left\| Lg_\phi(t, x_t) - \nabla_x g_\phi(t, x_t) \cdot v_t(x_t) \right\|^2. \quad (13)$$

The proof is provided in Appendix A. This result is key: it allows us to **optimize the correct marginal objective using the same efficient, simulation-free sampling strategy** as CFM training, bypassing the need to compute and store full ODE trajectories.

Moreover, this loss is a key distinction of our method. In contrast to one-step distillation methods, our approach is able to perform a true *linearization* of the **full dynamics**. The inclusion of the infinitesimal consistency loss, $\mathcal{L}_{\text{cons}}$, forces our Koopman representation to remain faithful to the teacher's dynamics **at every point along the trajectory**. We show a visual comparison of the trajectories, with and without $\mathcal{L}_{cons}$, in Figure 2, where the trajectories match the CFM trajectory *only* with the consistency loss.

**Unbiasedness, teacher fidelity, and diffusion.** The key ingredient of our method, Eq. 13, is an unbiased, simulation-free estimator of the marginal consistency loss (Prop. 3). As it is unbiased, any error in the learned $L$ and $g$ stems from

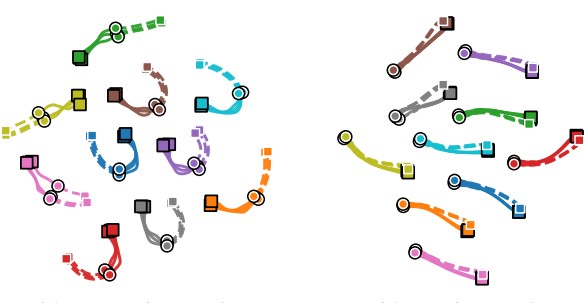

Without consistency loss      With consistency loss

*Figure 2.* t-SNE visualization of CFM and Koopman trajectories in the embedding space on FFHQ (test set). The consistency loss makes Koopman rollouts (dotted) follow the teacher dynamics (continuous) more closely. This is seen both in the proximity of trajectories and in the alignment of their endpoints. Circles mark starting points and squares mark end points.

the teacher's velocity field $v_t$, not from the linearization itself, and our machinery makes such discrepancies measurable (Sec. 6, *Insights on teacher training*). Moreover, nothing is specific to the precise CFM formulation: our proposition holds with any deterministic $v_t$ with evaluable marginals, including diffusion models via the probability-flow ODE (Song et al., 2021).

## 5. Global Linearization as an Interpretability and Control Tool

Our main goal is to expose an *interpretable* structure within generative dynamics by leveraging Koopman operator theory. Thanks to the mode decomposition of the operator L, our approach is a tool to shed light on the underlying dynamics and on the behavior of the teacher model.

### 5.1. Mode decomposition of Koopman operator

For this work, the main appeal of the Koopman theory is that it exposes an *interpretable* structure of the ODE, as we can decompose the different modes of the linear Koopman operator $L$. Intuitively, Koopman theory serves as a non-linear analogue of Fourier analysis: just as Fourier modes decompose signals into orthogonal oscillatory components, Koopman eigenfunctions decompose dynamics into independent modes with specific growth rates. We employ the *real Schur decomposition*:

$$L = QTQ^\top, \tag{14}$$

which represents each conjugate pair as a real $2 \times 2$ block and each real eigenvalue as a $1 \times 1$ block. A key property of the Koopman representation is that in Schur coordinates $y_t = Q^\top z_t$, the matrix exponential decomposes into independent modes. For a real eigenvalue $\lambda$, the corresponding $1 \times 1$ block yields an exponential mode $y(t) = e^{\lambda t} y(0)$, while

$2 \times 2$ blocks of the form

$$\begin{pmatrix} \sigma & \omega \\ -\omega & \sigma \end{pmatrix} \tag{15}$$

yield planar spirals $y(t) = e^{\sigma t} R(\omega t) y(0)$ with radial rate $\sigma$ and rotation frequency $\omega$. Importantly, in both cases, the norm of each component grows according to a predictable exponential rate: $e^{\lambda t}$ or $e^{\sigma t}$. This provides a canonical *ordering* to all the modes (akin to ordering Fourier modes by frequency). We show an image reconstruction with ordered modes in Figure 3.

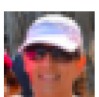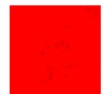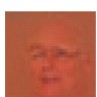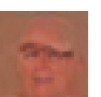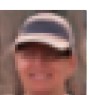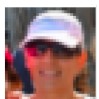

*Figure 3.* Progressive reconstruction of a test image as we progressively feed modes (N=400, 800, 100, full) with ascending real eigenvalue part. The modes appear to act in a coarse-to-fine manner

**Inversion of modes and images** A first step in interpreting Koopman modes is to "un-lift" them to the CFM dynamics. First, we highlight that our analytical sampling allows us to *invert any image $x$ into the noise space*, a task that is generally non-trivial for nonlinear generators and often requires specialized methods (e.g., (Mokady et al., 2023) for diffusion models). We do so by computing the corresponding latent noise $g(0, x_0) = \exp(-L)g(1, x)$ and optimizing noise in pixel space which reproduces the latent. We detail and demonstrate some inversion examples in the supplementary material F.1.

We can then unlift any mode $v_k$ into the pixel space. We do this by solving an inverse problem: Let $x_0 \sim p_0$ be a sample noise, $\mathbf{v}_i$ a Koopman mode. We search $x_{\text{pert}}^i$, such that:

$$x_{\text{pert}}^i = \arg\min_x ||g_\phi(0, x_0 + x) - g_\phi(0, x_0) + \alpha \mathbf{v}_i||^2. \tag{16}$$

This mode un-lifting is essential, as it allows to directly analyze the CFM teacher. We show an example of an unlifted mode in Fig. 4, compared with and without consistency. The model without consistency introduces notable noise artifacts.

### 5.2. Extended analysis of CFM models

Thanks to our global linearization, the spectral decomposition of $L$ (Section 5) is meaningful on the full trajectories, and the Koopman space is a *structured* latent space whose geometry is tied to the teacher's flow that we can explore with classical linear algebra. We use both to inspect three aspects of the CFM teacher: how generation is organized, whether the modes carry semantic content, and how the modes are discovered during training.

**Class-conditioned spectral signatures** Let a dataset $D = \{x_i\}$ and $D_c = \{x_i(c)\}$. We encode each image $x_i$ into its lifted representation $z_i$ and project it onto Koopman modes $(\phi_k)$, giving coefficients $\alpha_i(k) = |\langle\phi_k, z_i\rangle|$ and $\alpha_i(k, c) = |\langle\phi_k, z_i(c)\rangle|$. We then compute the dataset and class-averaged responses

$$\bar{\alpha}(k) = \frac{1}{|D|}\sum_{i \in D}\alpha_i(k), \bar{\alpha}(k, c) = \frac{1}{|D_c|}\sum_{i \in D_c}\alpha_i(k, c), \ (17)$$

and define the per-class transfer function

$$H(|\lambda|, c) = \bar{\alpha}(|\lambda|, c)/\bar{\alpha}(|\lambda|). \tag{18}$$

This measures how each class amplifies or suppresses modes of a spectral magnitude. By looking at which modes correspond to the highest class spectral deviation, we can understand which modes are common to images and which ones handle class-specific features.

**Semantic mode discovery** Given an image $x$, we perturb its lifted representation $z$ as $z'_k = z + \alpha v_k$. We measure the CLIP (Radford et al., 2021) - a common embedding space for text and images - similarity between the decoded image and some selected attribute $\beta$ prompts $p_\beta$. We define the *coherence* $C_k^\beta$ between a mode $v_k$ and an attribute $\beta$ as the sign consistency of similarity changes across test images:

$$C_k^\beta = \frac{1}{N}\sum_{i=1}^{N}\mathrm{sign}(\langle\mathrm{CLIP}(z'_k), \mathrm{CLIP}(p_\beta)\rangle - \langle\mathrm{CLIP}(z), \mathrm{CLIP}(p_\beta)\rangle)$$

$$(19)$$

We can then select modes $v_i$ with the highest coherence for different attributes, allowing semantic editing of the images, both in the Koopman space and in the un-lifted image space.

**Insights on Teacher Training** We use our Koopman framework to probe how the CFM teacher acquires its dynamics during training. We compare the modes $v_i^l, v_j^{\text{full}}$ at different training stages $l$ with the modes of the the fully trained teacher by computing their similarity $S_{ij}^{l,\text{full}}$, and further their cumulative similarity $c_s(k)$ to the full matrix $v_{\text{full}}$:

$$S_{ij}^{l,\text{full}} = \left|\langle(v_i^l)^\dagger v_j^{\text{full}}\rangle\right| / \left(\|v_i^l\|\|v_j^{\text{full}}\|\right), \ c_s(k) = \frac{1}{k}\sum_{i=1}^{k}S_{ii}^{l,\text{full}}$$

$$(20)$$

Those indicate how much the teacher has learned compared to the final teacher.

# 6. Experiments

To validate our framework, we investigate three key questions: (1) Can our one-step sampler achieve trajectory fidelity while keeping sampling performance? (2) Is the infinitesimal consistency loss ($\mathcal{L}_{cons}$) crucial for learning an interpretable linearization, as opposed to a simple boundary-matching distillation? (3) Does this learned structure lead

| Dataset | Mean MSE ($\times 10^{-5}$) |
|---|---|
| FFHQ (w/ consistency) | **0.5** |
| FFHQ (w/o consistency) | 130 |
| CIFAR-10 (w/ consistency) | **1.0** |
| CIFAR-10 (w/o consistency) | 174 |

*Table 1.* Average MSE between CFM trajectories and predicted Koopman trajectories. The consistency-trained model consistently outperforms the distilled model for trajectory fidelity.

to a more robust and functionally useful model? Our experiments show that only the model trained with $\mathcal{L}_{cons}$ learns a disentangled, editable, and robust generative process, while keeping a competitive generative quality relative to the purely distilled approaches or straight paths approaches (Appendix D.1). We highlight that generation quality is not our main goal and should be seen as a proxy for the capacity of our model to reproduce dynamics faithfully.

## 6.1. Experimental Setup

**Datasets and Teacher Model.** We evaluate mainly on CIFAR-10 (Krizhevsky, 2009) and a 32x32 downsampled version of the FFHQ face dataset (Karras et al., 2019). Our teacher is a pre-trained Optimal Transport Conditional Flow Matching (OT-CFM) model with a U-Net architecture. We highlight that our framework is teacher agnostic and not restricted to OT-CFM. For boundary-based losses ($\mathcal{L}_{\text{target}}$, $\mathcal{L}_{\text{phase}}$, $\mathcal{L}_{\text{recon}}$), we use 1 million pre-generated $(x_0, x_1)$ pairs from the teacher network.

**Koopman-CFM Architecture.** Our model consists of an encoder ($g_\phi$) and decoder ($g_\psi^{-1}$), both using a `SongUNet` architecture (Karras et al., 2022), which map to and from a 1024-dimensional latent space. The dynamics are governed by a learned affine linear generator ($\tilde{L}$). We provide a detailed breakdown of the computational complexity in Appendix B, Table 4. Notably, the exponentiation of the linear operator overhead remains negligible w.r.t the other operations.

**Training and Baselines.** We train for 800,000 iterations using the Adam optimizer (Kingma & Ba, 2015). Our primary baseline is an ablation of our own model trained without the consistency loss ($\mathcal{L}_{cons} = 0$), which reduces it to a standard distillation model, equivalent to (Berman et al., 2025).

## 6.2. Trajectory fidelity

We encode a teacher's trajectory $\{x_t\}_{t \in [0,1]}$ in the latent space and compare this ground truth path $z_t = g_\phi(t, x_t)$ against the analytical linear trajectory from our model, $\tilde{z}_t = \exp(\tilde{L}t)\tilde{z}_0$. We show the results in Table 1, with more results in the Appendix C.2. The trajectories are significantly better when using the consistency loss.

**No consistency**     **With consistency**

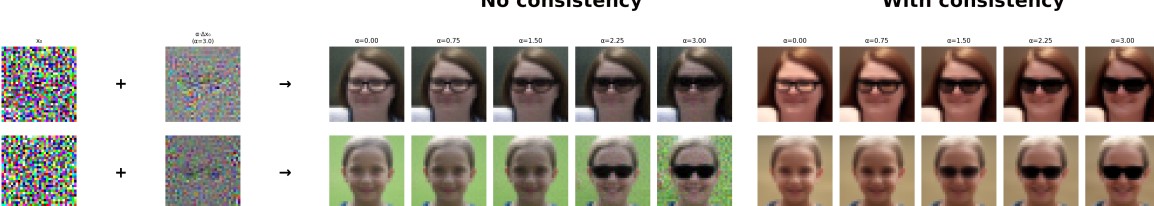

*Figure 4.* CFM-based semantic editing comparison. We port identified semantic directions from the Koopman latent space to the CFM noise-space via inversion as explained in F.3, here for sunglasses. Notably, we can see that recovered direction in the purely distilled model provides unreliable edits on generated images, as it comes with noticeable noisy artifacts, as opposed to our consistent model.

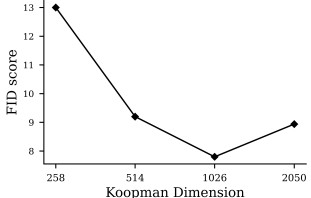

*Figure 5.* FID score as a function of Koopman dimension on the FFHQ dataset. The higher the dimension, the lower the FID.

| Attr./Coherence | W/ | | W/o | |
|---|---|---|---|---|
| | Max. | Var. | Max. | Var. |
| Glasses | **.97** | **.046** | .48 | .002 |
| Smile | .52 | **.010** | .50 | .003 |
| Brown | **.94** | **.027** | .50 | .003 |
| Young | **.45** | **.007** | .32 | .001 |
| Avg. | **.72** | **.022** | .45 | .002 |

*Table 2.* Semantic content of Koopman modes.: Maximal (Max.) and Variance (Var.) coherence of single-mode perturbations.

### 6.3. Ablation

**Koopman space dimension.** As shown in Figure 5, the Koopman dimension of 1026 (1024+2) is optimal for the generation quality. Notably, increasing the dimension to 1026 does not affect the quality with potential instabilities of the Koopman sampling components, such as the exponentiation. We also ablate the image dimension effect in Appendix C.3.

**Quality of the Koopman latent space.** We also provide a qualitative evaluation of the Koopman latent space. Namely, we borrow from the GAN literature and search for semantic directions in the latent space, such as glasses or gender. To find these directions in the latent space, we classify the dataset with attributes' CLIP ((Radford et al., 2021)) embedding similarity and compute the mean and difference with relevant latents, see Appendix F.2. for more details. Given a semantically coherent mode, we invert it to the image space and compare the quality of the semantic editing.

**Does a finite-dimensional Koopman representation exist?** Finite-dimensional Koopman representation is a strong condition that need not hold for arbitrary nonlinear flows (Iacob et al., 2023), so we cannot guarantee it *a priori*. Empirically, though, three observations hold jointly: competitive FID (Table 7), trajectory reproduction to within MSE $\sim 10^{-6}$ (Table 6 and Figure 2), and optimal FID value at an intermediary Koopman dimension as seen in Figure 5. Those observations suggest that a faithful global Koopman representation of the teacher exists in our learned coordi-

nates.

### 6.4. Interpretability Analysis

**Do Koopman Modes Encode Semantic Content?** We compute the coherence of modes on the FFHQ dataset with four attributes, namely, *glasses, smile, brown* and *young*. Table 2 compares the maximum coherence (Eq. 19) of models with and without consistency, as well as the maximum mean CLIP difference. The consistent model achieves near-perfect coherence for attributes like sunglasses (0.97) and brown hair (0.94), with variation magnitudes up to 24× larger. This demonstrates that consistency is essential for learning modes that align with interpretable semantics. We provide more results in the Appendix E.3.

**Can semantic directions transfer back to the original CFM?** Because the latent space is consistent with the teacher's flow, directions discovered in Koopman space are not decoder artifacts but meaningful perturbations of the CFM's original noise space. We use the approach described in Sec. F.3) to transfer semantic directions to the CFM noise space. This transfer succeeds only under consistency: otherwise, the back-mapped direction leaves the data manifold and CFM integration degrades and presents visible artifacts (Figs. 16, 17).

**How is the generation organized, what do the modes encode?** We show in Fig. 6 the different transfer functions (Eq. 18) for each class of CIFAR-10. Notably, low-energy modes are largely shared across classes, while higher-energy modes differentiate them. This is further supported by ob-

serving Figure 3, where the important elements of the face appear before differentiating ones, such as the smile, hat, and background.

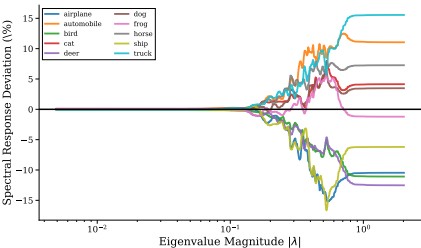

*Figure 6.* Per-class spectral deviation on CIFAR-10.

**How does the teacher acquire its dynamics during training?** Similarity matrices shown in Fig. 7 reveal a clear ordering: when modes are sorted by $\mathrm{Re}(\lambda)$, mid-training checkpoints already align with the low-decay modes of the final model, while early checkpoints show little correspondence. This is further quantified by the cumulative similarity, shown in Fig. 8, which increases monotonically across training stages.

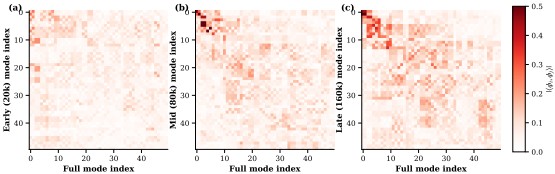

*Figure 7.* Eigenmodes similarity matrices comparing early and mid-training checkpoints against the fully trained model.

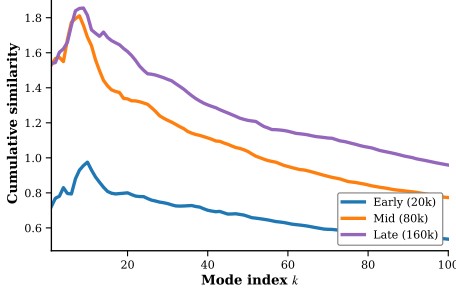

*Figure 8.* Cumulative average diagonal similarity shows progressive mode acquisition during training

**Outlook** The ability to pose these questions, and to answer them with linear algebra tools, rests on the non-trivial global linearization afforded by Koopman theory. We regard the present applications as an initial demonstration of what a faithful linear surrogate of a generative flow makes accessible, and as motivation for scaling such approaches to larger models and richer modalities.

## 7. Limitations

One remaining challenge is scaling our framework to high-resolution images, where the generator matrix becomes prohibitively large and its exponential can be numerically unstable. However, we note that modern generative models, including high-resolution DiTs, operate within a compressed latent space. Deploying our framework on these latent spaces presents no fundamental obstacles and is a highly promising direction for scaling. In the meantime, future work could explore structured operator approximations and specialized matrix exponential algorithms to address potential computational hurdles.

Another challenge is that we currently use pre-generated start-and-end pairs. While our unbiased estimator makes intermediate trajectory sampling simulation-free, avoiding the pair generation entirely is an open challenge, potentially solvable via novel operator regularization.

Furthermore, we observe that the quality gap between our method and traditional CFM widens on more complex datasets, motivating a deeper theoretical investigation into the conditions under which CFM dyamics admit a finite-dimensional Koopman representation (Iacob et al., 2023).

## 8. Conclusion

We introduced a principled Koopman operator framework to linearize Conditional Flow Matching, achieving fast, one-step, and interpretable generative modeling on realistic image domains, while uniquely linearizing the entire generation trajectory of the teacher model. Our novel consistency loss is key to enabling faithful reproduction of trajectories in the Koopman space and the decomposition of the Koopman matrix for practical interpretability applications. Moreover, our framework is independent of the velocity field and is not limited to flow matching. For example, diffusion SDEs can be formulated via the Fokker-Planck equation, which utilizes a velocity field that includes the score function. An exciting future work direction is to adapt our unbiased estimator for the consistency loss to enable similar applications on diffusion models. Finally, the modality-agnostic nature of our framework opens exciting avenues for adapting this linearization approach to other data types, such as audio and 3D shapes.

## Impact statement

As our method is primarily focused on advancing image generation, it raises similar concerns as for the potential misuse of generic image diffusion models for malicious applications (e.g., Deepfakes, impersonation) involving real individuals. To address this, it is essential to implement robust safeguards and ethical guidelines, similar to the se-

curity measures and NSFW content detection mechanisms already present in the existing commercial image generation pipelines.

## Acknowledgments

Parts of this work were supported by the ERC Consolidator Grant 101087347 (VEGA), as well as gifts from Ansys Inc., and Adobe Research.

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

# Appendix

The supplementary materials below provide an expanded theoretical motivation, experimental details, and additional results that support and extend the main paper. Each section corresponds to specific elements of our method and results, with backward references to the main paper for clarity.

## Appendix Overview

- **Section A: Theoretical Results and Proofs**
  In Section A we provide additional theoretical results and proofs, including the non-identifiability of Koopman coordinates, the non-equivalence of the conditional and marginal velocity field estimators and a tractable estimator for the marginal consistency loss.

- **Section B: Detailed Experimental Setup**
  In Section B we give details on the experimental setup, covering dataset preparation, architectures, hyperparameters, and computational resources.

- **Section C: Ablations**
  In Section C we present ablations, shedding light on the impact of loss terms on FID, the effect of Koopman dimension on FID, the role of consistency in trajectory fidelity, and the interpretability of modes with and without consistency.

- **Section D: Uncurated Samples and Sampling Speeds**
  In Section D we provide uncurated samples and wall-clock timings to further illustrate the speed–fidelity–interpretability tradeoff of our Koopman sampler.

- **Section E: Interpretability and Spectral Analysis**
  In Section E we analyze the learned Koopman representation, including mode structure with and without consistency, progressive coarse-to-fine reconstruction via eigenvalue ordering, class-conditioned spectral signatures, semantic mode discovery via CLIP coherence, and insights on teacher training dynamics.

- **Section F: Applications**
  In Section F we demonstrate practical applications of our framework, including image and mode inversion, discovering semantic directions in the Koopman latent space, noise engineering for CFM-based editing, and functional robustness on downstream tasks such as inpainting, super-resolution, and denoising.

- **Section G: Extended survey on interpretability of generative models**
  In Section G we provide a more extensive discussion on intepretrability of generative models.

Together, these sections provide a deeper understanding of our Koopman-CFM framework and support its efficiency, stability, and interpretability as claimed in the main paper.

## A. Theoretical Results and Proofs

In this section we expand on the theoretical foundations introduced in Section 4 of the main paper. We provide detailed proofs of Theorem 1 and Propositions 1–3, which establish the non-identifiability of Koopman coordinates up to linear transformations and justify the inclusion of the reconstruction loss, as well as the derivation of a tractable marginal consistency objective. These results complement the main text by giving formal guarantees for the claims underlying our Koopman-CFM framework.

### A.1. Preliminaries on CFM

We remind here the main components of Conditional Flow Matching (Tong et al., 2024), before deriving the proofs of our propositions. A Continuous Normalizing Flow (CNF) models the transformation from a prior distribution $p_0$ to a data distribution $p_1 = q_1$ via a probability path $p_t$. This path is induced by a time-dependent vector field $v_t$ through the ODE:

$$\frac{dx_t}{dt} = v_t(x_t), x_0 \sim p_0, x_1 \sim p_1 \tag{21}$$

where $x_t \in \mathbb{R}^d$ is a sample at time $t$. A naive objective to learn $v_t$ would be a regression loss:

$$\mathcal{L}_{\text{naive}} = \mathbb{E}_{t \sim U(0,1), x_t \sim p_t} \left\| v_\theta(t, x_t) - v_t(x_t) \right\|^2 \tag{22}$$

This objective is intractable because both the true vector field $v_t$ and the marginal path distribution $p_t$ are unknown. Conditional Flow Matching (CFM) circumvents this by defining a tractable conditional probability path $p_t(x_t|x_1)$ and its corresponding conditional vector field $u_t(x_t|x_1)$. The marginal velocity field $v_t$ can be expressed as an expectation over these conditional fields:

$$v_t(x_t) = \mathbb{E}_{x_1 \sim q(x_1|x_t)}[u_t(x_t|x_1)] = \int \frac{p_t(x_t|x_1)q(x_1)}{p_t(x_t)} u_t(x_t|x_1) dx_1 \tag{23}$$

Remarkably, CFM shows that minimizing a simulation-free objective based on the conditional velocity field is equivalent to minimizing the intractable marginal objective. The CFM loss is:

$$\mathcal{L}_{\text{CFM}} = \mathbb{E}_{t \sim U(0,1), x_1 \sim q_1, x_t \sim p_t(\cdot|x_1)} \left\| v_\theta(t, x_t) - u_t(x_t|x_1) \right\|^2 \tag{24}$$

While this makes training efficient, sampling requires solving the integral:

$$x_1 = x_0 + \int_0^1 v_\theta(s, x_s) ds \tag{25}$$

## A.2. Proof of Proposition 1

*Proof.* Let the augmented state observable be $E(t, x) = [t, g(t, x)]^T$. We show that the objectives are invariant under the transformation $E \mapsto E_T = T^{-1}E$ and $L \mapsto L_T = T^{-1}LT$ for any invertible block-diagonal matrix $T = \text{diag}(1, M)$.

We use two facts. First, the chain rule implies that the Jacobian transforms as:

$$\mathrm{D}(E_T)[1, v_t] = \mathrm{D}(T^{-1}E)[1, v_t] = T^{-1}\mathrm{D}E[1, v_t]. \tag{J}$$

Second, the matrix exponential (and thus the flow) is conjugate under $T$:

$$\exp(\Delta t L_T) = T^{-1} \exp(\Delta t L) T. \tag{C}$$

**Infinitesimal Consistency.** The residual is $R_{\text{cons}} = \mathrm{D}E[1, v_t] - LE$. The transformed residual is:

$$R_{\text{cons},T} = \mathrm{D}E_T[1, v_t] - L_T E_T \overset{(\mathrm{J}),(\mathrm{C})}{=} T^{-1}\mathrm{D}E[1, v_t] - T^{-1}LE = T^{-1}R_{\text{cons}}.$$

Thus, $R_{\text{cons}} = 0$ if and only if $R_{\text{cons},T} = 0$.

**Phase Loss.** The residual is $R_{\text{phase}} = E(1, x_1) - e^L E(0, x_0)$. The transformed residual is:

$$\begin{aligned} R_{\text{phase},T} &= E_T(1, x_1) - e^{L_T} E_T(0, x_0) \\ &= T^{-1}E(1, x_1) - (T^{-1}e^L T)(T^{-1}E(0, x_0)) \\ &= T^{-1}(E(1, x_1) - e^L E(0, x_0)) = T^{-1}R_{\text{phase}}. \end{aligned}$$

Again, the zero set of the loss is invariant. Since the norms of the residuals are scaled by the constant transformation $T^{-1}$, the set of global minimizers is preserved under this transformation. Therefore, the objectives only identify $g$ up to an invertible linear transformation $M$. $\square$

## A.3. Proof of Proposition 2

*Proof.* To simplify the notation, let us define:

$$\begin{aligned} A(x_t) &= Lg(x_t) \\ B(x_t) &= \nabla g(x_t)\, v_t(x_t) \\ C(x_t, x_1) &= \nabla g(x_t)\, u_t(x_t \mid x_1) \end{aligned}$$

With this notation, the losses are $\mathcal{L}_{\text{marg}} = \mathbb{E}_{x_t \sim p_t}[\|A(x_t) - B(x_t)\|^2]$ and $\mathcal{L}_{\text{cond}} = \mathbb{E}_{x_1 \sim q, x_t \sim p_t(\cdot|x_1)}[\|A(x_t) - C(x_t, x_1)\|^2]$.

We expand the squared norms inside the expectations:

$$\mathcal{L}_{\text{marg}} = \int p_t(x_t) \left( \|A\|^2 - 2\langle A, B \rangle + \|B\|^2 \right) dx_t$$

$$\mathcal{L}_{\text{cond}} = \iint q(x_1) p_t(x_t \mid x_1) \left( \|A\|^2 - 2\langle A, C \rangle + \|C\|^2 \right) dx_t \, dx_1$$

We will now compare the terms of these two expansions one by one.

**(i) First Term ($\|A\|^2$):** The first term of $\mathcal{L}_{\text{cond}}$ is $\iint q(x_1) p_t(x_t \mid x_1) \|A(x_t)\|^2 \, dx_t \, dx_1$. Since $A(x_t)$ does not depend on $x_1$, we can use the law of iterated expectation or simply rearrange the integral:

$$\iint q(x_1) p_t(x_t \mid x_1) \|A(x_t)\|^2 \, dx_t \, dx_1 = \int \left( \int q(x_1) p_t(x_t \mid x_1) \, dx_1 \right) \|A(x_t)\|^2 \, dx_t$$

$$= \int p_t(x_t) \|A(x_t)\|^2 \, dx_t$$

This is identical to the first term of $\mathcal{L}_{\text{marg}}$.

**(ii) Cross Term ($-2\langle A, \cdot \rangle$):** The cross term of $\mathcal{L}_{\text{cond}}$ is $\iint q(x_1) p_t(x_t \mid x_1) \left( -2\langle A(x_t), C(x_t, x_1) \rangle \right) dx_t \, dx_1$. We analyze the integral:

$$\iint q(x_1) p_t(x_t \mid x_1) \langle A(x_t), C(x_t, x_1) \rangle \, dx_t \, dx_1$$

$$= \int \left\langle A(x_t), \int q(x_1) p_t(x_t \mid x_1) C(x_t, x_1) \, dx_1 \right\rangle dx_t$$

$$= \int \left\langle A(x_t), \int q(x_1) p_t(x_t \mid x_1) \nabla g(x_t) u_t(x_t \mid x_1) \, dx_1 \right\rangle dx_t$$

$$= \int \left\langle A(x_t), \nabla g(x_t) \int q(x_1) p_t(x_t \mid x_1) u_t(x_t \mid x_1) \, dx_1 \right\rangle dx_t$$

By definition, the marginal velocity field $v_t(x_t)$ is the expectation of the conditional field $u_t(x_t \mid x_1)$ over the posterior $p(x_1 \mid x_t) = \frac{q(x_1) p_t(x_t|x_1)}{p_t(x_t)}$. So, $v_t(x_t) = \int u_t(x_t \mid x_1) \frac{q(x_1) p_t(x_t|x_1)}{p_t(x_t)} \, dx_1$. Multiplying by $p_t(x_t)$ gives $p_t(x_t) v_t(x_t) = \int q(x_1) p_t(x_t \mid x_1) u_t(x_t \mid x_1) \, dx_1$. Substituting this back into our expression:

$$\ldots = \int \langle A(x_t), \nabla g(x_t) \left( p_t(x_t) v_t(x_t) \right) \rangle \, dx_t$$

$$= \int \langle A(x_t), p_t(x_t) B(x_t) \rangle \, dx_t$$

$$= \int p_t(x_t) \langle A(x_t), B(x_t) \rangle \, dx_t$$

This shows that the cross terms of $\mathcal{L}_{\text{cond}}$ and $\mathcal{L}_{\text{marg}}$ are also identical.

**(iii) Final Quadratic Term ($\| \cdot \|^2$):** The final term of $\mathcal{L}_{\text{cond}}$ is $\mathbb{E}_{x_1, x_t}[\|C(x_t, x_1)\|^2]$. We use the law of total variance: for a random variable $Z$, $\mathbb{E}[\|Z\|^2] = \|\mathbb{E}[Z]\|^2 + \text{Var}(Z)$. We apply this by first conditioning on $x_t$.

$$\mathbb{E}_{x_1, x_t}[\|C\|^2] = \mathbb{E}_{x_t \sim p_t} \left[ \mathbb{E}_{x_1 \sim p(x_1|x_t)}[\|C(x_t, x_1)\|^2] \right]$$

$$= \mathbb{E}_{x_t} \left[ \|\mathbb{E}_{x_1|x_t}[C(x_t, x_1)]\|^2 + \text{Var}_{x_1|x_t}(C(x_t, x_1)) \right]$$

Let's compute the inner conditional expectation:

$$\mathbb{E}_{x_1|x_t}[C(x_t, x_1)] = \mathbb{E}_{x_1|x_t}[\nabla g(x_t) u_t(x_t \mid x_1)] = \nabla g(x_t) \mathbb{E}_{x_1|x_t}[u_t(x_t \mid x_1)] = \nabla g(x_t) v_t(x_t) = B(x_t).$$

Substituting this back:

$$\mathbb{E}_{x_1, x_t}[\|C\|^2] = \mathbb{E}_{x_t}\left[\|B(x_t)\|^2 + \text{Var}_{x_1|x_t}(C(x_t, x_1))\right]$$
$$= \mathbb{E}_{x_t}[\|B(x_t)\|^2] + \mathbb{E}_{x_t}[\text{Var}_{x_1|x_t}(C(x_t, x_1))]$$

The first part, $\mathbb{E}_{x_t}[\|B(x_t)\|^2] = \int p_t(x_t)\|B(x_t)\|^2\,dx_t$, is exactly the final term of $\mathcal{L}_{\text{marg}}$. The second part is the discrepancy term $\Delta(g)$:

$$\Delta(g) = \mathbb{E}_{x_t}[\text{Var}_{x_1|x_t}(C(x_t, x_1))]$$
$$= \mathbb{E}_{x_t}\left[\mathbb{E}_{x_1|x_t}\left[\|C(x_t, x_1) - \mathbb{E}_{x_1|x_t}[C(x_t, x_1)]\|^2\right]\right]$$
$$= \mathbb{E}_{x_t}\left[\mathbb{E}_{x_1|x_t}\left[\|C(x_t, x_1) - B(x_t)\|^2\right]\right]$$
$$= \mathbb{E}_{x_1, x_t}\left[\|C(x_t, x_1) - B(x_t)\|^2\right]$$
$$= \iint q(x_1)\, p_t(x_t \mid x_1)\|\nabla g(x_t)u_t(x_t \mid x_1) - \nabla g(x_t)v_t(x_t)\|^2\,dx_t\,dx_1$$
$$= \iint q(x_1)\, p_t(x_t \mid x_1)\|\nabla g(x_t)(u_t(x_t \mid x_1) - v_t(x_t))\|^2\,dx_t\,dx_1$$

**Conclusion:** Assembling all the terms, we have:

$$\mathcal{L}_{\text{cond}} = \underbrace{\mathbb{E}_{x_t}[\|A\|^2]}_{\text{Term 1}} - \underbrace{2\mathbb{E}_{x_t}[\langle A, B\rangle]}_{\text{Term 2}} + \underbrace{\left(\mathbb{E}_{x_t}[\|B\|^2] + \Delta(g)\right)}_{\text{Term 3}}$$
$$= \left(\mathbb{E}_{x_t}[\|A\|^2] - 2\mathbb{E}_{x_t}[\langle A, B\rangle] + \mathbb{E}_{x_t}[\|B\|^2]\right) + \Delta(g)$$
$$= \mathcal{L}_{\text{marg}} + \Delta(g)$$

Since $\Delta(g)$ is the expectation of a squared norm, it is non-negative, which proves the theorem. $\qquad\square$

### A.4. Proof of Proposition 3

*Proof.* The proof relies on the law of iterated expectation. Let $f(x_t)$ be any measurable function of $x_t$. The expectation of $f(x_t)$ over the marginal distribution $p_t(x_t)$ is:

$$\mathbb{E}_{x_t \sim p_t}[f(x_t)] = \int_{\mathbb{R}^d} f(x_t)p_t(x_t)\,dx_t$$

Now, we substitute the definition of the marginal path density, $p_t(x_t) = \int_{\mathbb{R}^d} q(x_1)p_t(x_t|x_1)\,dx_1$:

$$\mathbb{E}_{x_t \sim p_t}[f(x_t)] = \int_{\mathbb{R}^d} f(x_t)\left(\int_{\mathbb{R}^d} q(x_1)p_t(x_t|x_1)\,dx_1\right)dx_t$$

We can combine the terms inside a double integral:

$$\mathbb{E}_{x_t \sim p_t}[f(x_t)] = \iint_{\mathbb{R}^d \times \mathbb{R}^d} f(x_t)q(x_1)p_t(x_t|x_1)\,dx_1\,dx_t$$

By Fubini's theorem, we can exchange the order of integration since the integrand is non-negative (or integrable):

$$\mathbb{E}_{x_t \sim p_t}[f(x_t)] = \int_{\mathbb{R}^d} q(x_1)\left(\int_{\mathbb{R}^d} f(x_t)p_t(x_t|x_1)\,dx_t\right)dx_1$$

This expression can be recognized as a nested expectation. The inner integral is the expectation of $f(x_t)$ over the conditional distribution $p_t(\cdot|x_1)$, and the outer integral is the expectation over the data distribution $q(x_1)$:

$$\int_{\mathbb{R}^d} q(x_1)\left(\mathbb{E}_{x_t \sim p_t(\cdot|x_1)}[f(x_t)]\right)dx_1 = \mathbb{E}_{x_1 \sim q}\left[\mathbb{E}_{x_t \sim p_t(\cdot|x_1)}[f(x_t)]\right]$$
$$= \mathbb{E}_{x_1 \sim q, x_t \sim p_t(\cdot|x_1)}[f(x_t)]$$

---

**Algorithm 1** Koopman–CFM Training (simulation-free; fixed teacher, precomputed pairs)

---

**Input:** Fixed teacher velocity $v_{\text{CFM}}(t, x)$; encoder $g_\phi$; decoder $g_\psi^{-1}$; affine generator $\tilde{L}$; precomputed buffer $\mathcal{B} = \{(x_0, x_1)\}$.
**Definition:** Lifted coordinate $\tilde{z}(t, x) := [\, 1, \, t, \, g_\phi(t, x)\,]^\top$.
**for** each minibatch **do**

    Sample $x_1 \sim q_1, t \sim \mathcal{U}(0, 1)$, then draw $x_t \sim p_t(\cdot \mid x_1)$   $\mathcal{L}_{\text{cons}} \leftarrow \left\| \tilde{L}\, \tilde{z}(t, x_t) \;-\; Dg_\phi(t, x_t)[\, 1, \, v_{\text{CFM}}(t, x_t)\,] \right\|^2$

    Sample $(x_0, x_1)$ from buffer $\mathcal{B}$   $\mathcal{L}_{\text{phase}} \leftarrow \left\| \exp(\tilde{L})\, \tilde{z}(0, x_0) \;-\; \tilde{z}(1, x_1) \right\|^2$   $\mathcal{L}_{\text{target}} \leftarrow \ell_{\text{img}}\!\left( g_\psi^{-1}\!\left( \exp(\tilde{L})\, \tilde{z}(0, x_0)\right), \, x_1 \right)$

    $\mathcal{L}_{\text{recon}} \leftarrow \ell_{\text{img}}\!\left( g_\psi^{-1}\!\left( \tilde{z}(1, x_1)\right), \, x_1 \right)$

    $\mathcal{L} \leftarrow \lambda_c \mathcal{L}_{\text{cons}} + \lambda_p \mathcal{L}_{\text{phase}} + \lambda_t \mathcal{L}_{\text{target}} + \lambda_r \mathcal{L}_{\text{recon}}$   Update $\{\phi, \psi, \tilde{L}\}$ by backprop on $\mathcal{L}$
**end for**

---

**Algorithm 2** One-Step Koopman Sampling (matrix exponential + decode)

---

**Input:** Trained $(g_\phi, g_\psi^{-1}, L)$; prior $p_0 = \mathcal{N}(0, I)$.
**Input:** Lifted coordinate $z(t, x) := [\, 1, \, t, \, g_\phi(t, \, x)\,]^\top$. Precompute $E \leftarrow \exp(L)$   Sample $x_0 \sim p_0$
**Return:** $\hat{x}_1 \leftarrow g_\psi^{-1}\!\left( E\, z(0, \, x_0)\right)$

---

We have thus shown the general identity $\mathbb{E}_{x_t \sim p_t}[f(x_t)] = \mathbb{E}_{x_1 \sim q, x_t \sim p_t(\cdot \mid x_1)}[f(x_t)]$.

To prove the theorem, we simply choose $f(x_t)$ to be the squared residual of the marginal loss:

$$f(x_t) = \left\| \mathcal{L}g(x_t) - \nabla_x g(x_t)\, v_t(x_t) \right\|^2$$

By its definition, $\mathcal{L}_{marg} = \mathbb{E}_{x_t \sim p_t}[f(x_t)]$. Applying the identity we just derived gives:

$$\mathcal{L}_{marg} = \mathbb{E}_{x_1 \sim q, x_t \sim p_t(\cdot \mid x_1)} \left[ \left\| \mathcal{L}g(x_t) - \nabla_x g(x_t)\, v_t(x_t) \right\|^2 \right]$$

This completes the proof.     $\square$

## B. Experimental Details

This section complements Section 5 of the main paper by providing full details needed for reproducibility. We describe dataset prepration, model architecture and parametrization, training schedules, and computational resources.

**Data.** We evaluate our approach on three datasets of increasing difficulty. MNIST contains 60,000 training and 10,000 test grayscale images of handwritten digits at resolution $28 \times 28$. FFHQ (Flickr-Faces-HQ) was downscaled to $32 \times 32$ resolution, from which we use all 70,000 RGB images. Finally, CIFAR-10 provides 50,000 training and 10,000 test images at resolution $32 \times 32$ across 10 object classes. This progression from simple digits to natural faces and general object classes allows us to systematically study the performance of our method as task complexity increases.

**Model Architecture.** For all datasets, we employ a consistent backbone architecture: a SongUNet used as both encoder and decoder. To reduce the overall parameter count, we restrict the encoder output and decoder input to a single channel. Moreover, to obtain explicit control over the Koopman dimension, we optionally append a linear projection from the flattened UNet output to the target latent dimension.

**Training Details.** Before training our pipeline, we pre-trained an OT-CFM model following the reference implementation provided in the torchcfm[2] code examples. From this model, we generated between $10^4$ and $10^6$ $(x_0, x_1)$ pairs (see Table 3 for exact counts per dataset), which served as inputs for computing the target loss. All models were implemented in Pytorch (Ansel et al., 2024), trained using the Adam optimizer under identical training protocols across datasets. Experiments were carried out on NVIDIA A40, H100, and A100 GPUs. Additional hyperparameters, including learning rates, batch sizes, and training schedules, are reported in Table 3.

---

[2]https://github.com/atong01/conditional-flow-matching

| | MNIST | FFHQ | CIFAR-10 |
|---|---|---|---|
| CFM iterations | 200k | 800k | 800k |
| Batch size | 128 | 256 | 124 |
| Learning rate | 0.0001 | 0.0001 | 0.0001 |
| Koopman iterations | 70k | 600k | 800k |
| Target weight (w/o $\mathcal{L}_{\text{cons}}$ – w/ $\mathcal{L}_{\text{cons}}$) | 1.0 – 1.0 | 1.0 – 0.01 | 1.0 – 0.01 |
| Operator Dimension | 1026 | 1026 | 1026 |
| UNet Output Channels | 1 | 1 | 1 |
| UNet Base Channels | 64 | 64 | 64 |
| UNet Channels Multiplier | [1,2,2] | [1,2,2,2] | [1,2,2,2] |
| Linear Projection | ✓ | ✗ | ✗ |

*Table 3.* Training hyperparameters for Koopman–CFM on MNIST, FFHQ, and CIFAR-10. *Linear projection* refers to the projection head at the UNet encoder output (resp. decoder input). Since loss terms are not of the same order of magnitude, the target loss was reweighted by the given parameter. *Koopman iterations* denote the number of iterations for the overall pipeline, while *CFM iterations* correspond to the underlying CFM model.

*Table 4.* Latency breakdown across pipeline stages.

| Stage | Latency (ms) |
|---|---|
| Encoding | $19.058 \pm 1.126$ |
| Latent evolution w/o expm | $0.023 \pm 0.001$ |
| Latent evolution w/ expm | $1.211 \pm 0.012$ |
| Decoding | $18.513 \pm 0.597$ |
| Everything w/o expm | $36.631 \pm 0.329$ |
| Everything w/ expm | $38.838 \pm 0.316$ |

**Computational complexity.** We show in Table 4 the decomposition of computation times of our approach. Notably, the computation budget is mostly spent on the U-Net Encoding and Decoding operations. The matrix exponential is computed at each iteration using PyTorch's matrix_exp (scaling and squaring with Padé approximants (Bader et al., 2019)). With a moderate Koopman dimension (1026), it remains negligible.

## C. Ablations

This section expands the analysis of Section 5 by presenting ablations that clarify the role of each loss term, the effect of Koopman dimension, the impact of consistency on trajectory fidelity, and the interpretability of modes.

### C.1. Impact of Loss Terms

Table 5 shows the effect of adding loss components across datasets. Phase and reconstruction alone yield poor FIDs, as they impose no constraint in image space. Adding the target loss improves fidelity by supervising decoded samples. Adding the consistency loss (weight 0.01) slightly worsens FID (e.g., FFHQ $7.5 \rightarrow 10.1$), since it regularizes the model to follow the teacher's nonlinear trajectories rather than shortcutting through straighter ones. This increases trajectory faithfulness at the cost of marginally higher endpoint error. We argue this tradeoff is beneficial: while endpoint-only distillation can optimize FID, it fails to capture the true generative flow (cf. Table 6, Fig. 2). Consistency-trained models achieve competitive FIDs while uniquely enabling spectral decomposition and robust downstream performance.

Since we're optimizing a composite loss, there may be concerns of instability during training. For transparency we provide plots of the behavior of all our loss terms. Training is stable, and can be rationalized with well aligned objectives derived from both Koopman theory and the CFM framework.

*Table 5.* Loss ablation across datasets showing the effect of incrementally adding loss components

| Dataset | $\mathcal{L}_{\text{recon}} + \mathcal{L}_{\text{phase}}$ | $\mathcal{L}_{\text{recon}} + \mathcal{L}_{\text{phase}} + \mathcal{L}_{\text{target}}$ | $\mathcal{L}_{\text{recon}} + \mathcal{L}_{\text{phase}} + 0.01\mathcal{L}_{\text{target}} + \mathcal{L}_{\text{cons}}$ |
|---|---|---|---|
| MNIST | 143.5 | 6.43 | 7.1 |
| FFHQ | 41 | 7.5 | 10.1 |
| CIFAR-10 | 64.5 | 14.1 | 16.7 |

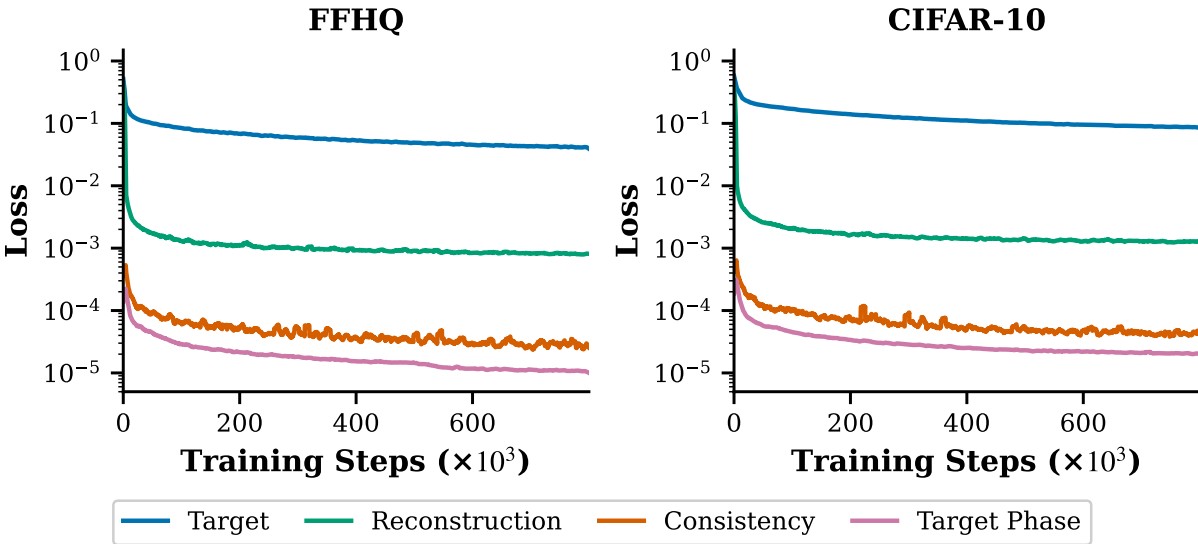

*Figure 9.* Training losses on CIFAR-10 and FFHQ datasets across different loss components.

## C.2. Trajectory Fidelity with and without Consistency Loss

To visualize trajectory fidelity in an interpretable coordinate system, we project dynamics onto the Schur basis of the learned generator $L$ (Figure 10). Each $2 \times 2$ block corresponds to a complex eigenvalue pair $\sigma \pm i\omega$, where $\sigma$ governs the exponential envelope and $\omega$ the oscillation frequency. With consistency training, the learned Koopman modes accurately track the CFM teacher's trajectory throughout, confirming that the representation captures the true generative dynamics. Without consistency, the endpoints remain correct, explaining the comparable generation quality, but the intermediate trajectory diverges significantly from the teacher. This demonstrates that consistency loss is essential for learning Koopman representations whose modes faithfully reflect the underlying flow, rather than merely learning a shortcut between boundaries.

## C.3. Dimension scalability

Results in Table 7 show that we can learn Koopman representations with increasing variability, from the relatively simple MNIST to more complex datasets like FFHQ and CIFAR-10. To assess the dimension scalability, we trained a Koopman generator on FFHQ images of dimension 64x64. We obtain a FID of **13.4**, showing similar results when the dimension increases.

*Table 6.* MSE between trajectory rollouts between CFM and Koopman dynamics in latent space: we generate 1000 full trajectories $\{x_t\}$ via CFM encode in the Koopman latent space $g(t, x_t)$ and compare them with Koopman rollouts $g(x_t) = e^{Lt}g(t = 0, x_0)$.

| Dataset - Values ($\times 10^{-6}$) | Min | Max | Mean MSE | Std Dev |
|---|---|---|---|---|
| FFHQ (w/ consistency) | **3.0** | **13** | **5.0** | **1.0** |
| FFHQ (w/o consistency) | 524 | 2660 | 1300 | 295 |
| CIFAR-10 (w/ consistency) | **4.0** | **37** | **10** | **4.0** |
| CIFAR-10 (w/o consistency) | 346 | 7010 | 1740 | 636 |

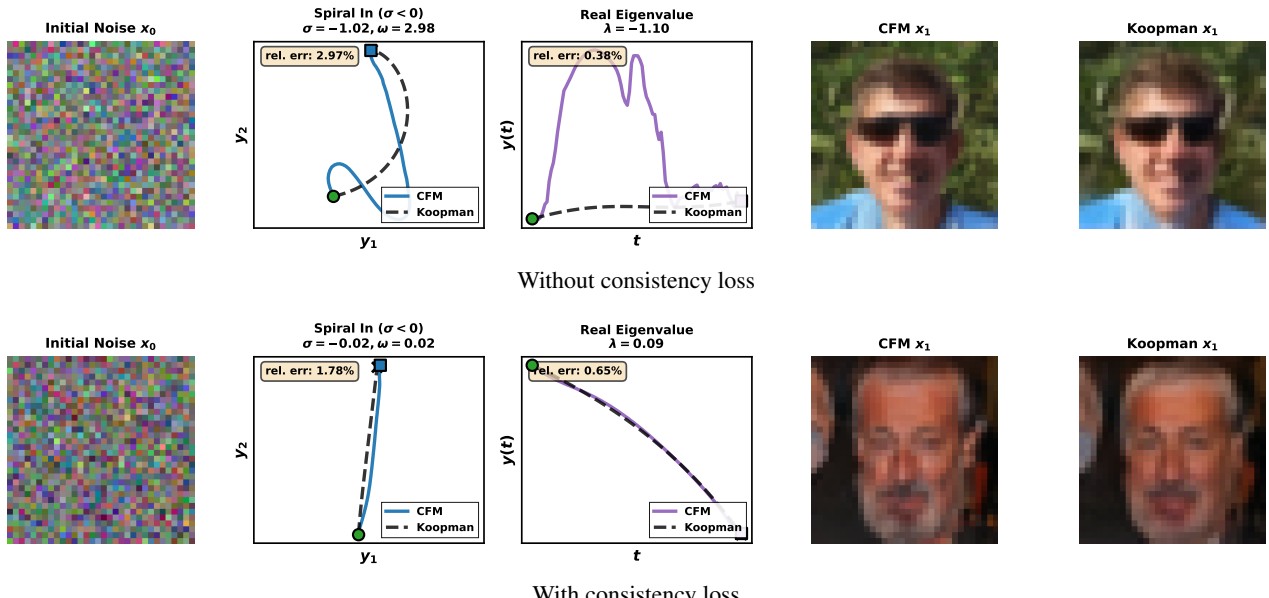

*Figure 10.* Trajectory comparison in Schur coordinates of the learned Koopman generator. With consistency, learned Koopman modes (dashed) accurately track CFM dynamics (solid). Without consistency, endpoints match but intermediate trajectories diverge, indicating the learned modes do not reflect true CFM dynamics.

## D. Sampling with Koopman operator

This section supplements Section 5 by showing generation quality, uncurated generations and reporting wall-clock sampling times, illustrating the tradeoffs between, speed, fidelity and interpretability.

### D.1. Generation Quality

*Table 7.* FID (↓) and sampling time (s/img, ↓) on three benchmark datasets. Our Koopman formulation achieves competitive or superior generation quality while enabling fast inference. Baselines are trained under identical preprocessing for fair comparisons. ♯ Indicates reproduction. Rectified Flow uses 2RF training and 1-step distillation.

| Method | NFE | MNIST | FFHQ | CIFAR-10 | Sampling Time (ms/img) |
|---|---|---|---|---|---|
| Koopman (ours, w/ consistency) | 1 | 7.1 | 10.1 | 16.7 | 37.2 |
| Koopman (ours, w/o consistency) | 1 | 6.4 | 7.5 | 14.1 | 37.2 |
| OT-CFM | 1 | 181 | 149 | 226 | 7.1 |
| OT-CFM | 3 | 28.1 | 51 | 59.3 | 25.2 |
| OT-CFM | 5 | 12.5 | 31.4 | 31.5 | 41.1 |
| OT-CFM | 25 | 4.4 | 11.6 | 12.3 | 209 |
| OT-CFM ((Tong et al., 2024)) | 100 | 1.9 | 8.5 | 7 | 849 |
| Rectified Flow ((Liu et al., 2023)) | 1 | ♯ 1.76 | ♯ 4.23 | 4.85 | ♯ 24.9 |
| MeanFlow ((Geng et al., 2025)) | 1 | ♯ 4.03 | ♯ 3.34 | ♯ 3.59 | ♯ 22.5 |

We evaluate sample quality using the Fréchet Inception Distance (FID) (Heusel et al., 2017), shown in Table 7, on MNIST (LeCun et al., 2010), FFHQ, and CIFAR-10. Our full Koopman-CFM model with consistency achieves competitive performance. Interestingly, the model trained without consistency achieves a slightly superior FID on FFHQ (7.5 vs. 8.5 for the teacher). This suggests that, when only constraining the endpoints, the distillation model is free to find a combination of paths and latent space that is easier to learn. As mentioned above, however, such a model is not guaranteed to replicate the trajectories of the teacher model. We provide uncurated generated examples with the consistency trained model in the appendix Section D.

### D.2. Uncurated samples

Uncurated samples on MNIST, FFHQ and CIFAR-10 are shown in Figure 11.

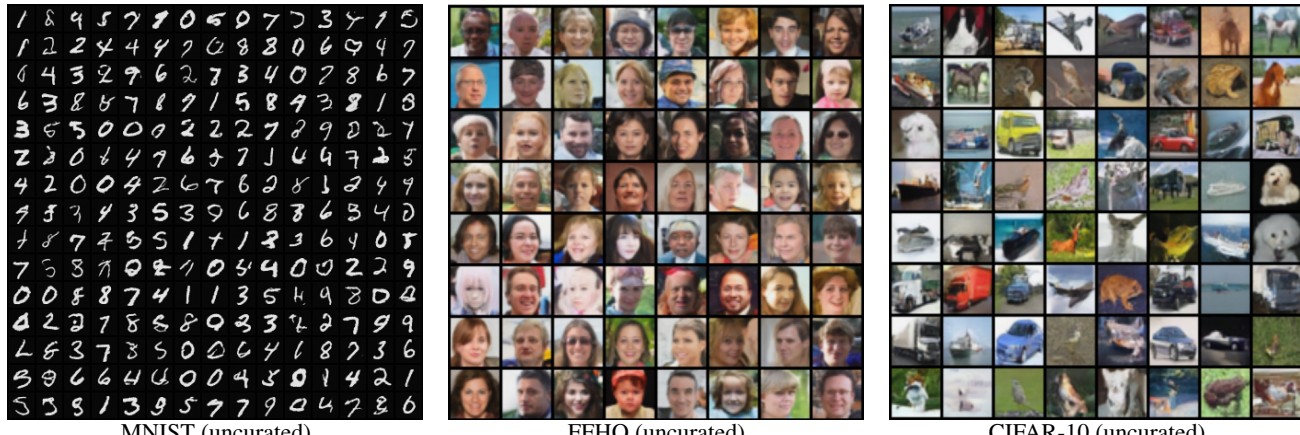

MNIST (uncurated)       FFHQ (uncurated)       CIFAR-10 (uncurated)

*Figure 11.* Uncurated samples from our Koopman generative model across three datasets. All samples are obtained via our one-step strategy.

## E. Interpretability and Spectral Analysis

### E.1. Koopman mode structure

Figure 12 illustrates how consistency qualitatively changes the learned Koopman modes. Without consistency, individual modes tend to decode into entire faces—effectively full puzzle pieces—which suggests poor disentanglement, as each mode redundantly encodes the whole sample. By contrast, with consistency, the modes behave like localized "patch bases," decomposing faces into local interpretable components close to semantic components (e.g., hair, eyes). The spectral profile on the left of Fig. 12 also highlights important differences: with consistency, coefficients decay with eigenvalue magnitude, whereas without consistency the spectrum remains flat, indicating the absence of structured decomposition.

### E.2. Generation process: coarse-to-fine.

To investigate the interpretability of the learned Koopman representation, we perform progressive mode reconstruction by truncating the eigenspectrum of the generator $L$. Specifically, we compute the eigendecomposition of the feature block $A_{gg} = L_{[2:,2:]}$ and construct a real-valued basis by taking $\mathrm{Re}(v)$ and $\mathrm{Im}(v)$ for each complex conjugate eigenvector pair. We sort modes by the real part of their eigenvalues, $\mathrm{Re}(\lambda)$, which governs the exponential timescale of each mode: more negative values correspond to strongly decaying dynamics while values closer to zero or positive correspond to slowly decaying or amplifying dynamics.

Given an encoded image $z = [1, t, g]^\top$ where $g \in \mathbb{R}^{1024}$ denotes the feature vector, we reconstruct using only the first $k$ modes by projecting onto the truncated basis $B_k \in \mathbb{R}^{1024 \times k}$:

$$\hat{g}_k = B_k B_k^\dagger g, \tag{26}$$

where $B_k^\dagger$ denotes the pseudoinverse. The reconstructed features $\hat{g}_k$ are then decoded back to image space.

Figure 13 shows reconstructions for increasing $k$. The slowest modes ($k \leq 200$, $\mathrm{Re}(\lambda) \leq -0.58$) produce homogenous outputs, indicating these modes encode a global bias that requires additional modes to balance. At $k = 400$, coarse facial structure emerges, face shape, average skin tone, and approximate feature positions. As $k$ increases to 600–800, identity-specific features begin to appear, though images remain soft. Finally, modes with $\mathrm{Re}(\lambda) > 0$ ($k > 1000$) contribute fine details: hair texture, sharp edges, and accessories such as hats and glasses. Full reconstruction recovers the original image with high fidelity.

These results demonstrate that the Koopman eigenspectrum induces a principled coarse-to-fine hierarchy: slow modes capture global structure while fast modes encode high-frequency details. Unlike PCA, which orders components by variance,

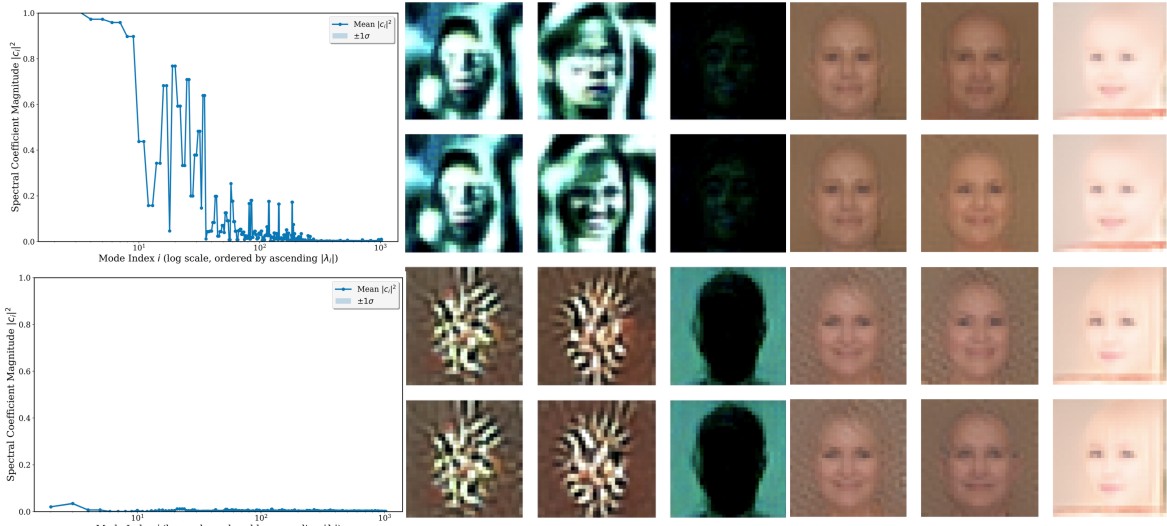

*Figure 12.* Left: Mean coefficients $|c_i|^2$ projected on the generator modes ordered by corresponding eigenvalue magnitude $|\lambda_i|^2$. Top corresponds to the spectrum along the modes obtained from training with consistency and bottom to those obtained from training without consistency Right: First three columns are some decoded modes of the generator trained with consistency loss, and the next three are those obtained from the generator trained without consistency.

this ordering emerges from the *dynamics* of the generative flow, providing an interpretable decomposition tied to the underlying generative process.

### E.3. Koopman Modes Align with Semantic Directions

We provide uncurated qualitative examples of Koopman mode-induced image edits in Figure 14.a, Figure 14.c, Figure 14.d, Figure 14.b. Each row shows a different test image, with columns corresponding to perturbation strengths $\alpha \in \{-0.2, -0.1, 0, 0.1, 0.2\}$. Importantly, these modes were not manually selected; rather, they were automatically identified by ranking all eigenmodes according to their CLIP coherence scores with respect to each attribute prompt.

**Sunglasses (Mode 1019).** For the model trained with consistency loss, this mode demonstrates strong semantic alignment: positive $\alpha$ consistently introduces sunglasses across diverse subjects while preserving identity, pose, and background. Negative $\alpha$ produces the inverse effect, brightening the eye region and removing eyewear. The transformation generalizes across ages, genders, and lighting conditions, confirming that this eigenmode captures a disentangled semantic direction rather than spurious correlations.

**Brown Hair (Mode 767).** This mode exhibits coupling between hair color and global illumination. While positive $\alpha$ shifts toward darker hair tones, it simultaneously reduces overall image brightness. This entanglement suggests that some semantic attributes share spectral structure in the Koopman operator, consistent with the lower selectivity scores reported in Table **??**.

**Effect of Consistency Loss.** Without consistency loss, perturbing Koopman modes produces no discernible change in the decoded images, regardless of the perturbation magnitude $\alpha$. In contrast, the consistency-trained model yields clearly visible and semantically coherent edits. This qualitative difference corroborates the quantitative findings in Section 6.4: consistency loss is essential not only for learning a faithful linear decomposition of the dynamics, but also for ensuring that the resulting eigenmodes correspond to actionable semantic directions in image space.

## F. Applications

### F.1. Inversion

To edit a real image $\mathbf{x}$, we first invert it to a corresponding noise sample $\mathbf{x}_0$ such that integrating the CFM ODE recovers the original image. We encode the image to its lifted representation $\mathbf{z} = g(\mathbf{x})$ and compute the target noise-space embedding

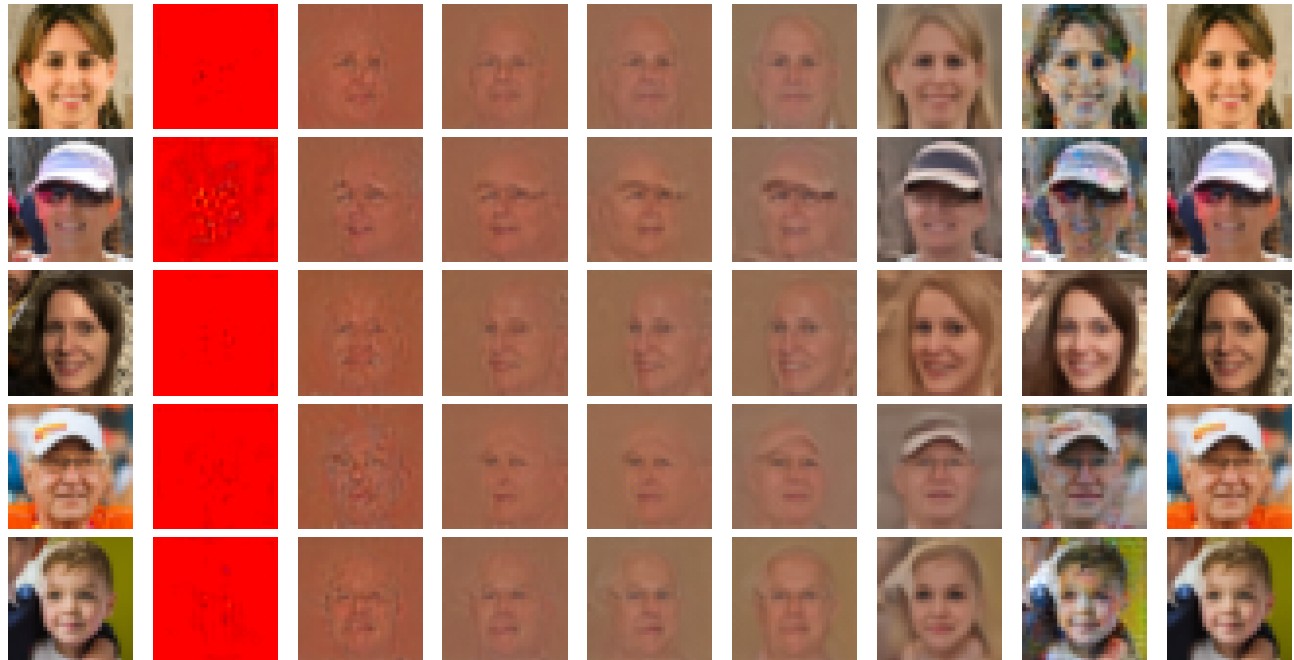

*Figure 13.* Progressive mode reconstruction sorted by $\mathrm{Re}(\lambda)$. Slow modes (negative $\mathrm{Re}(\lambda)$) capture coarse structure, while fast modes (positive $\mathrm{Re}(\lambda)$) add fine details. The learned Koopman spectrum provides an interpretable hierarchy reflecting the multi-scale nature of the generative process.

$\mathbf{z}_0^* = \exp(-L)\mathbf{z}_1$. We then optimize $\mathbf{x}_0$ to match this target:

$$\mathbf{x}_0^* = \arg\min_{\mathbf{x}_0} \|g(\mathbf{x}_0) - \mathbf{z}_0^*\|_2^2 \tag{27}$$

We show inversion examples in Figure 15. We again highlight the difference with the no-consistency model, which introduces artifacts and have noticeable unstable optimization.

### F.2. Discovering semantic directions in Koopman latent space

We discover semantic directions in the lifted Koopman space using a supervised approach. Given the FFHQ dataset, we first encode each image $\mathbf{x}$ into its lifted representation

$$\mathbf{z} = g(\mathbf{x})$$

at $t = 1$. Binary attribute labels (e.g., smiling vs. not smiling, eyeglasses vs. no eyeglasses) are obtained via CLIP classification using natural-language prompts.

For each binary attribute, we compute a semantic direction as the difference between class-conditional mean embeddings:

$$\mathbf{d}_{\mathrm{attr}} = \mathbb{E}[\mathbf{z} \mid y = 1] - \mathbb{E}[\mathbf{z} \mid y = 0]. \tag{28}$$

Semantic editing is performed via linear traversal in the Koopman latent space:

$$\mathbf{z}_{\mathrm{edited}} = \mathbf{z} + \alpha\, \mathbf{d}_{\mathrm{attr}}, \tag{29}$$

where $\alpha$ controls the edit strength. The edited latent code is then decoded back to image space via

$$\hat{\mathbf{x}} = g^{-1}(\mathbf{z}_{\mathrm{edited}}).$$

We show, in Table 8, that the provided latent directions are better with the consistency model, both in Koopman or unlifted to the image space, with CLIP and LPIPS evaluations for different attributes.

| | No Consistency | | Consistency | |
|---|---|---|---|---|
| Direction | CLIP ↑ | LPIPS ↓ | CLIP ↑ | LPIPS ↓ |
| *Direct Koopman Editing* | | | | |
| Hat | 0.069 | **0.018** | **0.077** | 0.019 |
| Sunglasses | 0.070 | 0.035 | **0.072** | **0.034** |
| Smile | **0.093** | **0.016** | 0.090 | 0.020 |
| Age | 0.121 | **0.035** | **0.123** | 0.036 |
| Gender (M→W) | 0.153 | **0.021** | **0.157** | 0.025 |
| Gender (W→M) | 0.150 | **0.031** | **0.152** | 0.035 |
| *CFM-Based Editing* | | | | |
| Hat | **0.148** | 0.060 | 0.094 | **0.053** |
| Sunglasses | **0.130** | 0.076 | 0.107 | **0.056** |
| Smile | **0.087** | 0.013 | 0.052 | **0.010** |
| Age | **0.186** | 0.097 | 0.145 | **0.087** |
| Gender | **0.178** | 0.091 | 0.139 | **0.043** |

*Table 8.* Quantitative comparison of latent directions, at $\alpha = 3.0$

### F.3. Noise engineering: Performing image editing by optimizing CFM noise perturbation

A key advantage of our Koopman-based framework is that semantic directions discovered in the lifted space can be transferred to perform editing with the original CFM model. This demonstrates that the Koopman operator captures meaningful structure that generalizes beyond the learned decoder.

**Optimizing semantic perturbations.** Rather than directly adding $\mathbf{d}_{\text{attr}}^{(0)}$ in pixel space, we optimize a perturbation $\Delta \mathbf{x}_0$ such that the perturbed noise induces the desired semantic shift in the lifted space:

$$\Delta \mathbf{x}_0^* = \arg\min_{\Delta \mathbf{x}_0} \left\| g(\mathbf{x}_0 + \Delta \mathbf{x}_0) - \left( g(\mathbf{x}_0) + \mathbf{d}_{\text{attr}}^{(0)} \right) \right\|_2^2 \tag{30}$$

Edited images are then generated by integrating the perturbed noise through the *original* CFM model:

$$\hat{\mathbf{x}}_1 = \int_0^1 v_\theta(t, \, \mathbf{x}_0 + \alpha \, \Delta \mathbf{x}_0^*) \, dt \tag{31}$$

where $\alpha$ controls the edit strength and $v_\theta$ is the pretrained CFM velocity field.

**Role of consistency regularization.** We observe a stark difference in editing quality depending on whether the Koopman model was trained with consistency loss. Figure 16 and Figure 17 compares semantic traversals for models trained with and without this loss term.

With consistency regularization, the optimized perturbations remain well-behaved across a wide range of edit strengths ($\alpha \in [0, 3]$). Edits are semantically meaningful, identity is preserved, and image quality remains stable even at large $\alpha$ values. In contrast, without consistency regularization, edited images exhibit severe degradation at moderate-to-large perturbation strengths: backgrounds become corrupted with color artifacts, facial structure deteriorates, and identity is lost.

We attribute this to the role of consistency loss in aligning the Koopman dynamics with the underlying CFM trajectory. When this alignment is enforced, the learned operator $\exp(L)$ accurately models how features evolve under the flow, ensuring that mapped directions $\mathbf{d}_{\text{attr}}^{(0)} = \mathbf{d}_{\text{attr}} \exp(-L)$ correspond to valid perturbations within the noise distribution's support. Without this constraint, the backward mapping may produce directions that push samples off the data manifold, causing the CFM integration to generate out-of-distribution outputs.

These results highlight that our Koopman framework not only enables direct editing via the learned decoder $g^{-1}$, but also provides a principled mechanism for *noise engineering*, transferring semantic control to any compatible generative model by operating in its noise space.

### F.4. Functional Robustness on Downstream Tasks

Finally, we evaluate if this interpretable structure of our framework translates to challenging downstream tasks: inpainting, super-resolution, and denoising. These tasks test the model's ability to perform conditional generation, which depends on the quality of its learned dynamics. For a corrupted input encoded to $z_{1,corr}$, we reconstruct by adding noise at $t = 0$ and evolving it through the learned process:

$$z_{0,corr} = e^{-L} z_{1,corr} \quad ; \quad x_{recon} = g_\psi^{-1}(e^L(z_{0,corr} + \text{noise}))$$

As shown in Figure 18, the consistency-trained model significantly outperforms the ablation model across all tasks. This superior performance is a direct consequence of the structured, Fourier-like basis described above. Because its learned dynamics can induce local, patch-based semantic modifications, the model is uniquely equipped to solve tasks that require local reasoning, like inpainting a missing patch. The purely distilled model fails and simply reproduces the same image, showing that it only learned the noise-to-data map, instead of the underlying image data distribution.

## G. Extended survey on interpretability of generative models

There is a rich body of work on understanding how generative models transform noise into data. Early research on VAEs and GANs focused on analyzing their latent spaces. Variational Autoencoders were used to learn *disentangled* representations of data (Bengio et al., 2013), i.e., latent codes that separate the underlying generative factors of variation (Higgins et al., 2017; Burgess et al., 2017; Kim & Mnih, 2018; Khemakhem et al., 2020). The success of Generative Adversarial Networks (Goodfellow et al., 2014) prompted similar studies (Chen et al., 2016). Because the latent space of GANs is not explicitly structured, research focused on identifying directions that correspond to interpretable generative factors, enabling controlled image editing (Jahanian et al., 2020; Härkönen et al., 2020; Voynov & Babenko, 2020; Shen & Zhou, 2021). The rise of diffusion and flow models as state-of-the-art generative methods naturally raised the question of whether such interpretability techniques could be extended to these models. However, their iterative generation process and the prevalence of complex, learnable control mechanisms (Zhang et al., 2023) have not yielded equally simple or powerful methods for interpretation and editing. Existing approaches tend to be more complicated and lack the conceptual clarity and usability of those developed for VAEs and GANs (Kwon et al., 2023; Yang et al., 2023; Meng et al., 2022; Kulikov et al., 2025). In contrast, our method preserves the dynamical-systems view of these models while enabling simple and interpretable latent-space manipulations.

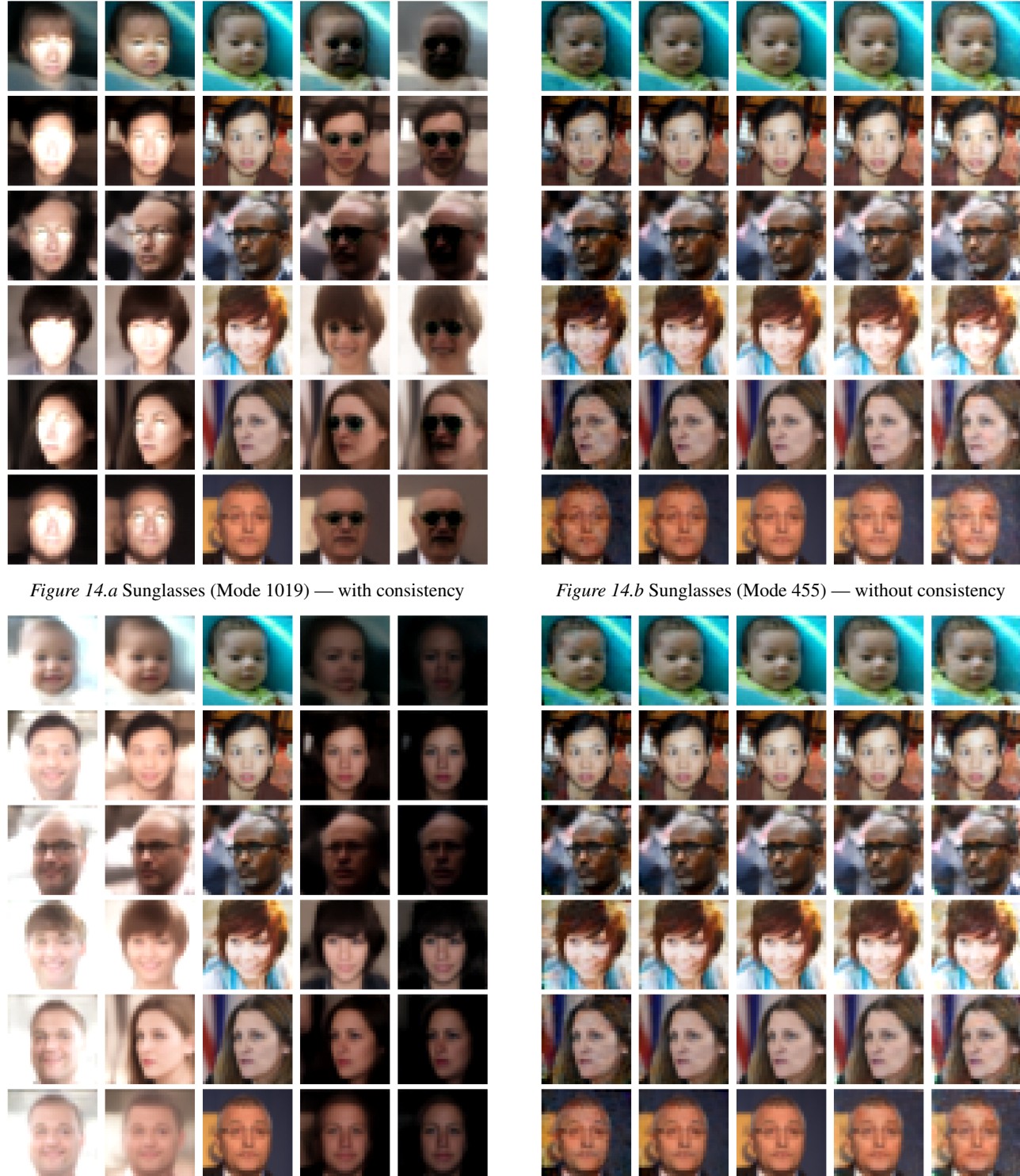

*Figure 14.a* Sunglasses (Mode 1019) — with consistency

*Figure 14.b* Sunglasses (Mode 455) — without consistency

*Figure 14.c* Brown hair (Mode 767) — with consistency

*Figure 14.d* Brown hair (Mode 2) — without consistency

*Figure 14.* **Koopman mode perturbations for semantic editing.** Each grid shows different test subjects (rows) perturbed with $\alpha \in \{-0.2, -0.1, 0, 0.1, 0.2\}$ (columns). Modes were automatically identified via CLIP coherence analysis. **(a, c)** With consistency loss, perturbing individual eigenmodes produces semantically meaningful edits—adding sunglasses or darkening hair—while preserving identity. **(b, d)** Without consistency loss, the same perturbations yield no visible change, demonstrating that consistency training is essential for learning actionable semantic directions.

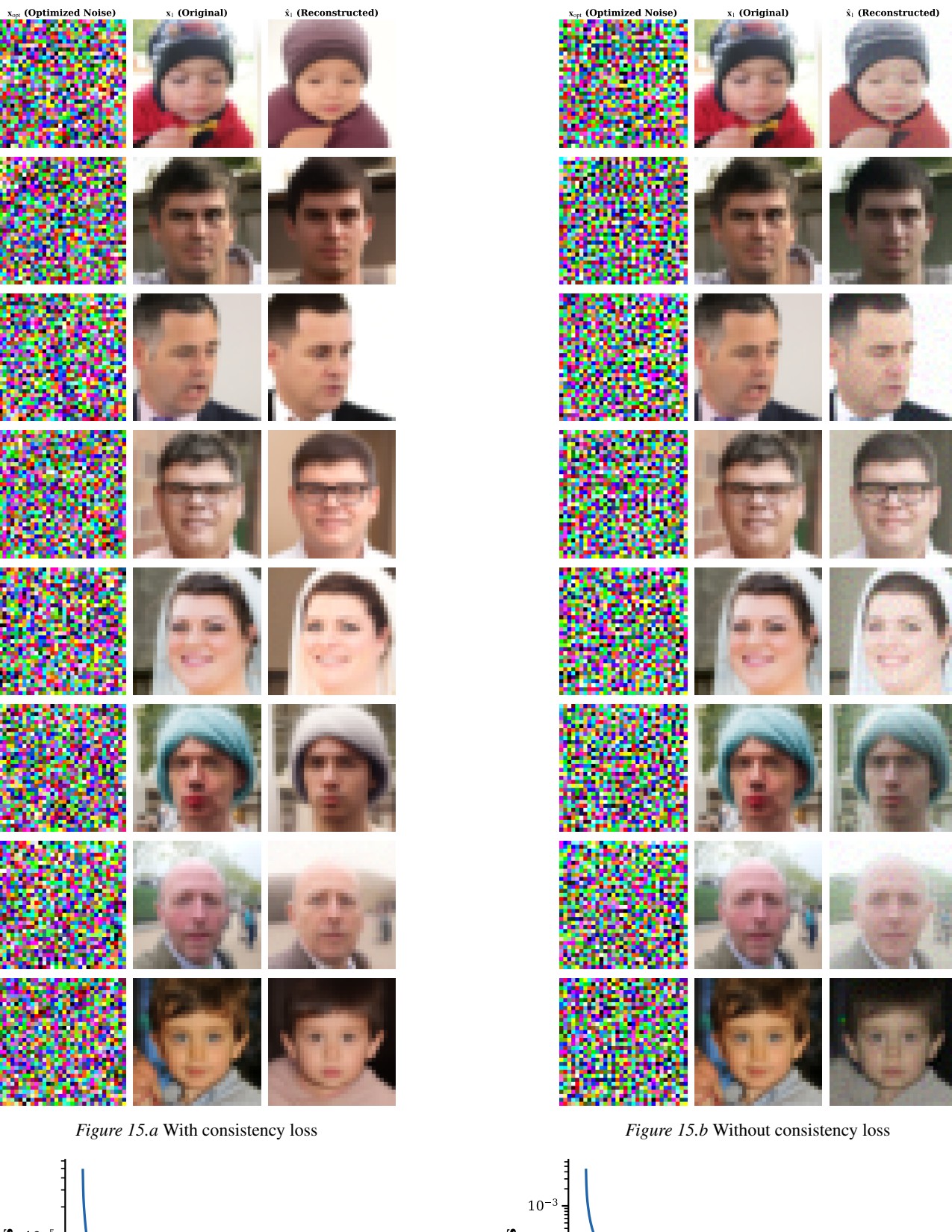

Figure 15.a With consistency loss

Figure 15.b Without consistency loss

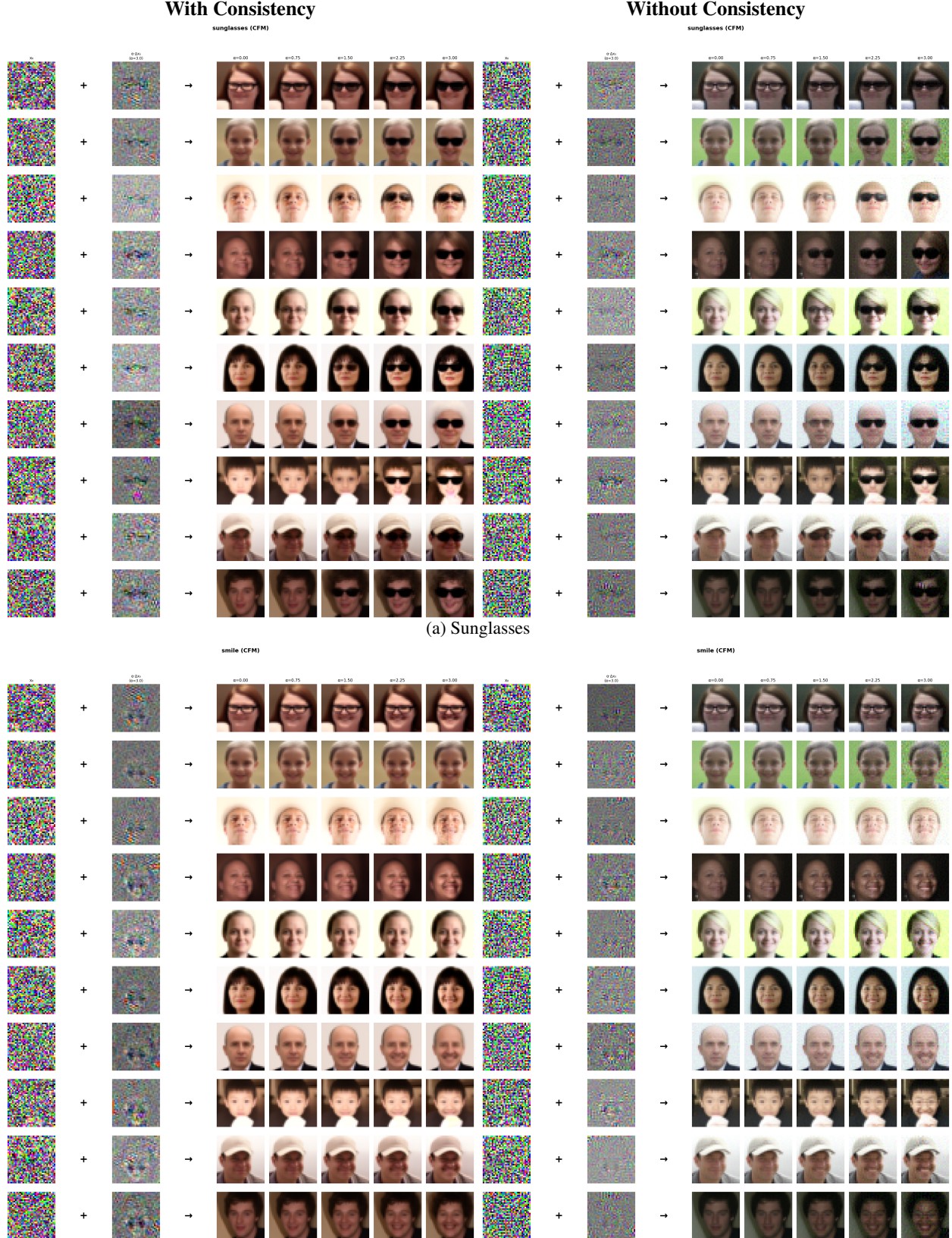

*Figure 16.* **Consistency loss enables stable semantic editing.** Edits via optimized noise perturbations $\mathbf{x}_0 + \alpha\Delta\mathbf{x}_0$ integrated through CFM ($\alpha \in [0, 3]$, increasing left-to-right). With consistency loss (left), edits remain coherent and identity-preserving. Without (right), large $\alpha$ causes artifacts and structural collapse.

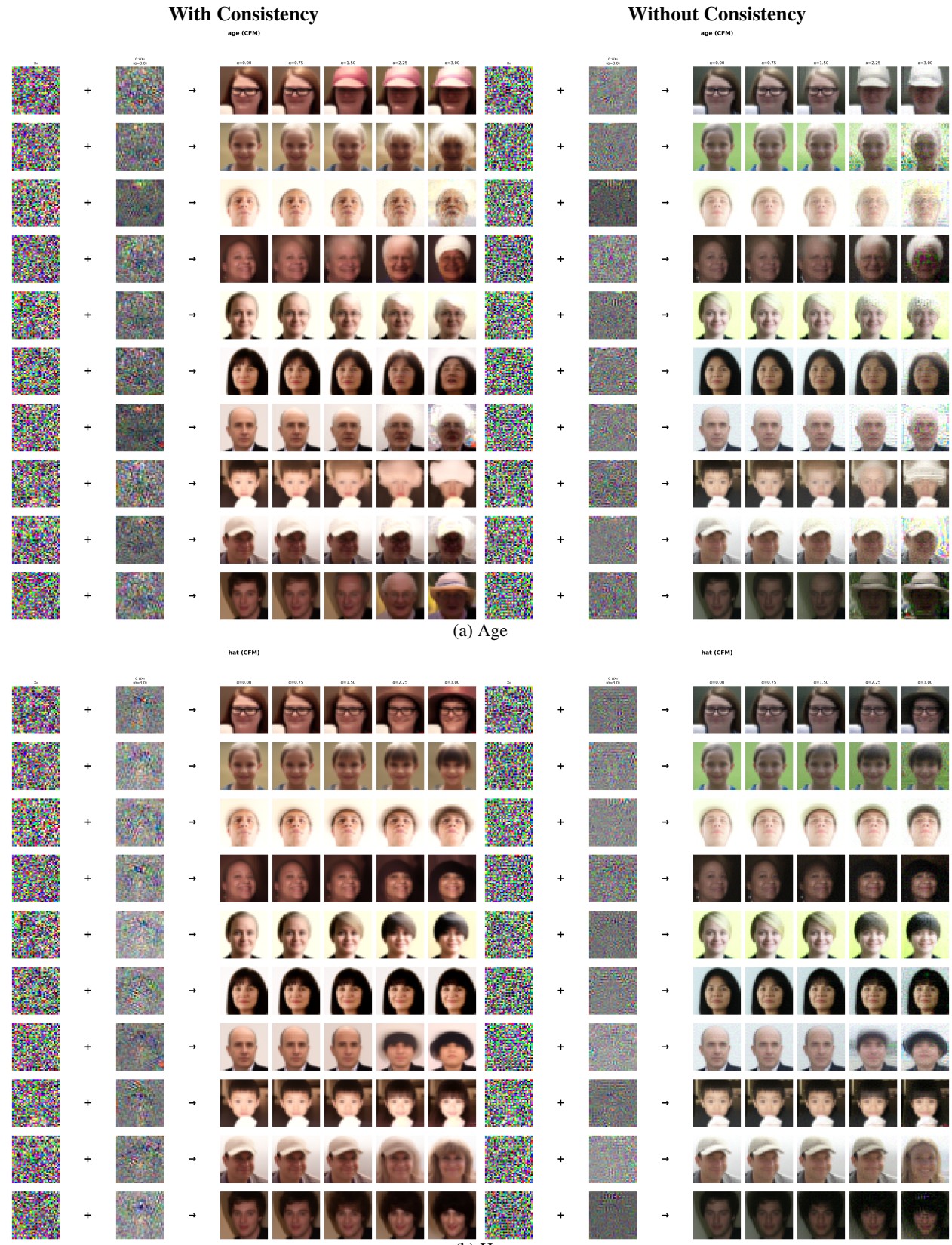

*Figure 17.* **Additional semantic directions.** Same setup as Figure 16. Consistency loss enables stable traversal across diverse attributes.

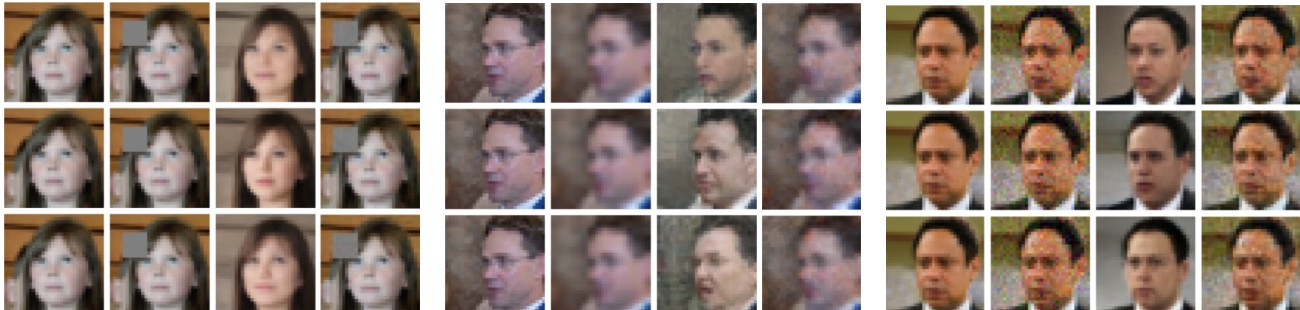

*Figure 18.a* Inpainting          *Figure 18.b* Super-Resolution          *Figure 18.c* Denoising

*Figure 18.* Performance on structured generative tasks. For each task, we show the input, the corrupted image, the result from our consistency-trained model, and the result from the ablation model. Each row corresponds to the application of different gaussian noise. Our model consistently produces coherent, high-fidelity results, while the ablation model fails.

