# OpenReview forum: "Unfolding Generative Flows with Koopman Operators: Trajectory-Preserving Linearization"
_ICML.cc/2026/Conference — ICML 2026 regular_

### Official Review · Reviewer_aM7t · 2026-03-06

**Soundness:** 3
**Presentation:** 2
**Significance:** 3
**Originality:** 3
**Overall Recommendation:** 5
**Confidence:** 3

**Summary:**

The authors propose a method to distill flow matching models while projecting the latent space onto a learned space in which a Koopman operator allows for end-to-end linear and thus one-step generation.
A new "consistency" loss component is introduced which encourages an encoder to find embeddings in which the teacher's trajectories can be linearized with a jointly learned Koopman operator.
The composition of the learned operator and opportunities for interpretability and control of the model is studied.
The authors validate that, with this additional loss,
1) the rolled-out linear trajectories in the learned lifted space are more faithful to the flow matching trajectories encoded at each step than without.
2) the modes of the learned operator align more closely with image semantics than without
3) the quality of semantically edited images is higher than without

**Compliance With Llm Reviewing Policy:**

Affirmed.

**Final Justification:**

In my view issues with the presentation have been largely addressed. The main limitations that remain are:

- Scaling to high image dimensions may be possible, but is unclear
- The consistency loss presents a trade-off between generative quality and trajectory linearization

As stated in my review, interpretability of multistep generative models is a hard task and I am aware of little progress made in this direction. Therefore I think this work, contributing a possible pathway for interpretability, should be published despite these limitations. Within the scope of the paper, the authors have shown the merit of their novel consistency loss, and it is appropriate to delegate addressing the remaining limitations to future work in my opinion. Therefore I vote to accept this paper.

**Key Questions For Authors:**

Q1: Regarding Editing (Appendix F.3, Eq. 31). Why is the perturbed image $x_0 + \alpha \Delta x_0^*$ integrated using the original CFM model and not the Koopman-based model?
Q2: How does including $\mathcal L_\text{recon}$ solve the non-identifiability problem? With a learnable encoder-decoder pair, if $\mathcal L_\text{phase}$ and $\mathcal L_\text{target}$ are satisfied as in Prop 1, in what scenario can $\mathcal L_\text{recon}$ still be non-zero at t=1? Is this related to the mention of a fixed decoder in Corollary 1.1?
Q3: Could you provide a rough contribution of the computation of $e^L$ versus $g, g^{-1}$ to the training and inference time in your experiments?
Q4: Regarding Section 6.3 and Figure 5. It is claimed in Section 6.3 that the optimal Koopman dimension is 1024+2. Does "following the trend" of Figure 5 not suggest further FID improvements beyond the claimed optimal dimension of 1024+2? Did you stop at 1026 due to computational constraints?
Q5: Regarding Section 5.2, Insights on Teacher Training: How meaningful is the comparison of these modes? Should / can we expect a "hot diagonal" for asymptotically long training in Figure 7?

**Limitations:**

yes

**Strengths And Weaknesses:**

## Soundness
The proposed method is based on established Koopman operator theory and recent flow matching literature.
Koopman-CFM distillation provides a novel interpretability tool for trained CFM models that can be trained conveniently post-hoc.
The authors provide convincing evidence of their claims in forms of figures and tables (exception: see Q4).
Common evaluation scores (FID) are used to assess generation quality.
The appended proofs appear correct (but have not been checked in detail).
This approach requires learning the high-dimensional matrix $L$ and the use of the matrix exponential (or Schur decomposition), which has complexity $d^3$. To assess the severity of this limitation could you provide a rough contribution of the computation of $e^{tL}$ versus $g, g^{-1}$ to the training and inference time in your experiments?

## Presentation
The main text is fairly well written and structured, and the relation to prior work made very clear.
The results come a bit short in the main text.

Critique and Suggestions (in no particular order):

General:
- Readers may not immediately recognize that the work addresses distillation exclusively. This should be made clearer, especially early on.
- I find section 5 hard to understand at several points without being already familiar with Koopman theory. Please consider extending it or adding a deeper explanation of the concepts to the appendix for unfamiliar readers
- Please specify (preferably in the captions) where the data is used for your tables and figures comes from: train / test / something else? (Figure 2, Figure 3, Figure 4, Table 1).

Explicit:
- Equation for $L$ should be mentioned in the main text rather in Appendix B.1. Is it correct that the "1" entry is not on the diagonal?
- Eq 3: Please address why this result is significant and what is is used for (implicitly or explicitly) in the work.
- Some figures are tiny or low-resolution (e.g. 6, 7, 8)
- The time dependence of $v$ and $g$ is not made explicit in section 3.2, which leads to inconsistent notation to the later sections. Please consider introducing a consistent, time-parameterized notation at the start of the paper.
- Eq 19 $\mathcal{v}^{l, \dagger}$ should be $({\mathcal{v}^l})^{\dagger}$?
- Figure 3: Images are not annotated as to the number of modes used to create them
- Table 1: Caption claims mean and std but table only shows mean MSE. Please include both or change caption. Also, setting the scale to MSE (in $10^{-6}$) to eliminate the scientific notation would look a bit nicer
- Collary 1.1: mentiones a fixed decoder, but decoder is learned. Is this intended?
- The tables in the Appendix would benefit from highlighting the best value in bold.
- Section 2, Interpreting and Explaining Generative Models: First sentence is incomplete.
- Appendix B.2: Please specify the approximate number of parameters of the models.
- Why do the results of table 4 and 6 not match? Did you use different $\lambda$?


## Significance
The work addresses a relevant topic: single-step generation and interpretability of generative continuous flow-based models.
While the generation quality and sampling time of the proposed method are currently well below state-of-the-art methods, it offers the added benefit of interpretability.
This is a hard task for multistep generative models where the trajectories are fixed by the forward ODE/SDE.
Tables 4 & 6 suggest that the consistency loss is counterproductive for FID, which presents a trade-off between interpretability and generation quality. This indicates that the main usecase of Koopman-CFM is likely not speeding up generation, but providing the benefits that come with linearized trajectories, which is accurately identified by the authors.
Even though the method is currently limited to relatively small images, this is an important step towards interpretable flow matching models.

## Originality
The proposed method is similar to the cited prior work Berman et al. 2025. "One-Step Offline Distillation of Diffusion-based Models via Koopman Modeling".
The novelty lies within the theoretical and experimental validation of a new consistency loss, which the authors show brings benefits of improved interpretability, sample inversion and editing.

---

> ### Author Rebuttal · Authors · 2026-03-31
>
> Dear Reviewer aM7t,
>
> We sincerely thank you for your thoughtful review. We are pleased that you found our application of Koopman operator theory well-founded, our writing clear, and our interpretability results compelling. As you correctly identified, while there is currently an FID trade-off, our primary objective is to unlock the "black box" of continuous generative models and provide mathematically grounded interpretability and control.
>
> Below, we address your specific questions and critiques:
>
> ## Questions
>
> **Q3: Computational complexity.**
> The Schur decomposition is $O(d^3)$ but performed once post-training. The matrix exponential is computed at each iteration using PyTorch’s matrix_exp (scaling and squaring with Padé approximants [1]). With a moderate Koopman dimension (1026), it remains negligible compared to U-Net cost (see table).
>
> | Stage                     | Latency (ms)   |
> | ------------------------- | -------------- |
> | Encoding                  | 19.058 ± 1.126 |
> | Latent evolution w/o expm | 0.023 ± 0.001  |
> | Latent evolution w/ expm  | 1.211 ± 0.012  |
> | Decoding                  | 18.513 ± 0.597 |
> | Everything w/o expm       | 36.631 ± 0.329 |
> | Everything w/ expm        | 38.838 ± 0.316 |
>
> *Sampling latency breakdown for the Koopman-CFM CIFAR10 pipeline. Inputs are
> 3×32×32. Values are reported as mean ± std, on 16 warmup runs, 100 timed runs. w/o expm is used at inference time, where we store $e^L$ once and use it at each generation.*
>
>
> **Q1**
> By mapping Koopman semantic directions back to the CFM noise space, we show they remain faithful to the original model. This "best of both worlds", hybrid approach combines precise linear control in Koopman space with the teacher’s output quality, avoiding degradation from the distilled decoder.
>
> **Q2: Non-identifiability, Corollary 1.1.**
> Proposition 1 highlights that Koopman observables in functional space have a linear degree of freedom, which hinders invertibility. The $L_{recon}$ term acts as a soft constraint to explicitly enforce this invertibility by selecting a decodable coordinate system from the equivalence class of valid Koopman representations. The “fixed decoder” in Corollary 1.1 was a typo; the result holds with a jointly learned decoder.
>
> **Q4**
> To test this, we trained the pipeline with a Koopman dimension of 2048+2. We observed that FID performance did not improve noticeably (FID of *8.94*). This, coupled with the fact that our linearized model at d=1026 captures the teacher's FID, suggests that the non-linear dynamics are well-captured at this dimension.
>
> **Q5**
> This comparison reveals spectral learning biases in Flow Matching. Sorting modes by real part provides a consistent ordering, showing that low-frequency (global) modes are learned first, followed by high-frequency details. Extending across checkpoints would likely yield an asymptotic hot diagonal.
>
> ## Typos and Suggestions
>
> Thanks for your meticulous reading of the manuscript! We will implement all of your suggestions/corrections in the final version.
>
> 1. **Framing, Exposition:** We will clarify the presentation. Our method distills a CFM teacher using continuous Koopman theory of dynamics systems. It differs from pure distillation, as spectral tools unlock a range of interpretability analysis on the teacher. We will provide a more intuitive, step-by-step breakdown of Koopman theory in Section 5.
> 2. **Figure Captions:** The data for Figures 2, 3, 4, and Table 1 come from the test split. We will specify this in the final version.
> 3. **Operator L Equation:** We will move the parameterization of L from the Appendix to the main text.  '1' is indeed off-diagonal to explicitly enforce the time evolution constraint ṫ=1.
> 4. **Equation/Proposition 3:** We will clarify the text to emphasize that Proposition 3 (the unbiased estimator) is the bedrock of our pipeline, as it enables the simulation-free training that makes this method tractable.
> 5. **Figure Quality and Annotations:** We will generate high-resolution versions of Figures 6, 7, and 8. In Figure 3, the number of modes are 400, 800, 1000, and full.
> 6. **Time Notation:** We will introduce a consistent, time notation for $v,g$ early in Section 3.2 to prevent later confusion.
> 7. **Equation 19 Typo:** We will correct to $(v^l)^†$.
> 8. **Tables:** Table 1 stds are in Table 5, sorry for the mistake. We will update Table 1, remove scientific notation, and bold the best values in all tables.
> 9. **Section 2:** We will correct the sentence, as the point was meant to be a comma.
> 10. **Parameter count:** We will include them in Appendix B.2.
> 11. **Tables Mismatch:** The mismatch in Table 6 is an accidental carryover from an older experiment. We will match it to Table 4.
>
> [1] "Bader, P.; Blanes, S.; Casas, F. "Computing the Matrix Exponential with an Optimized Taylor Polynomial Approximation." Mathematics 2019

---

> > ### Author Rebuttal · Reviewer_aM7t · 2026-04-01
> >
> > Thank you for addressing my concerns. I encourage you to include the experiments conducted for Q3 and Q4 in the paper.
> >
> > In my view issues with the presentation (reviewers yQSA, DZKj, aM7t) have been largely addressed. The main limitations that remain are:
> > - Scaling to high image dimensions may be possible, but is unclear (reviewers yQSA, 9oGs, aM7t)
> > - The consistency loss presents a trade-off between generative quality and trajectory linearization (reviewers 9oGs, DZKj, aM7t)
> >
> > As stated in my review, interpretability of multistep generative models is a hard task and I am aware of little progress made in this direction. Therefore I think this work, contributing a possible pathway for interpretability, should be published despite these limitations. Within the scope of the paper, the authors have shown the merit of their novel consistency loss, and it is appropriate to delegate addressing the remaining limitations to future work in my opinion. I will raise my score accordingly.

---

### Official Review · Reviewer_DZKj · 2026-03-10

**Soundness:** 3
**Presentation:** 2
**Significance:** 4
**Originality:** 4
**Overall Recommendation:** 4
**Confidence:** 4

**Summary:**

The paper applies Koopman operator theory to flow matching / diffusion models to analyze the generative process. This entails learning a finite-dimensional representation of each step of the diffusion process using an encoder-decoder network and encouraging linearity in the diffusion dynamics in this space through the objective functions. The authors then use linear analytical methods on the learnt representations to interpret a pre-trained diffusion model.

**Compliance With Llm Reviewing Policy:**

Affirmed.

**Final Justification:**

The authors addressed all my concerns and I have raised my confidence from 3→4 of my positive score. I agree with the other reviewers' concerns regarding scalability but believe the merits of this work outweigh this weakness.

**Key Questions For Authors:**

1. How can you be sure that the finite-dimensional Koopman representations are capturing the complete nonlinear dynamics? Without confirming this, does it invalidate the linear interpretability analysis?
2. Are you able to match the teacher FID score at high enough Koopman dimension?

**Limitations:**

I discussed limitations above.

**Strengths And Weaknesses:**

**Soundness**

Strengths:

- Central idea of applying Koopman operator theory to flow matching for interpretability makes sense.
- The key novelty with respect to Berman et al. 2025, the trajectory consistency loss, is well supported theoretically and appears to be effective in ablation studies.
- Interpretability studies are broadly reasonable and very interesting.

Weaknesses:

- Central assumption that there exists a finite-dimensional Koopman representation is not supported. Rather, it is contradicted by poor FID scores in Fig. 5 (though the teacher FID score is not given as a point of comparison). Therefore, it is unclear how valid linear interpretability methods are.
- Koopman dimension study in Fig. 5 stops at 1026D. Would be useful to know where the performance saturates, and how it compares to the teacher FID score.
- Inversion results (Fig. 15) are not convincing; in fact, the “without consistency loss” reconstructions visually appear closer to the original image.

**Significance and originality**

As authors acknowledge, the idea is closely related to that of Berman et al. 2025; however, I believe the continuous-time formulation and the trajectory consistency loss are significant and original contributions demonstrated by the authors on a range of creative interpretability studies. I particularly liked the progressive reconstruction and semantic editing studies.

**Presentation**

There are a few sloppy or incorrect equations and statements, which are unfortunately distracting and hinder the legibility of the paper. Some examples: Eq. 2: x_0 ~ p_0, not g(x_0); MeanFlow (Geng et al. 2025) is described as “endpoint pair regression”, which is incorrect; phi_k in Sec. 5.2 is undefined; incorrect references e.g. to App B in Sec 4.3 and labels of propositions in Appendix.

---

> ### Author Rebuttal · Authors · 2026-03-31
>
> Dear Reviewer DZKj,
>
> We thank you for your thoughtful review. We are happy that you found the central idea of applying Koopman operator theory to flow matching theoretically sound and that our "key novelty ..., the trajectory consistency loss,  is well supported theoretically and appears to be effective in ablation studies." We also appreciate your validation of our interpretability studies as "broadly reasonable and very interesting", particularly the progressive reconstruction and semantic editing.
>
> Below, we address your main questions and concerns:
>
> **Finite-Dimensionality and Koopman Dimension Ablation (W1, W2, Q1, Q2)**
>
> You correctly pointed out that assuming a finite-dimensional Koopman representation is a strong claim, and questioned whether stopping at a dimension of 1026 simply reflected computational limits.
>
> To directly address this, we conducted an additional ablation where we trained our Koopman pipeline with a dimension of 2048+2. We observed that FID performance did not improve noticeably (FID of *8.9* on FFHQ). This suggests that the non-linear dynamics are **well-captured at this dimension**.
>
> Furthermore, we respectfully push back on the characterization of the FID scores as "poor". For the FFHQ dataset, our model trained without consistency actually _matches or slightly beats the teacher's FID_ (*7.5* vs *8.5* (teacher)), and remains highly competitive when the consistency loss is applied. Given that our method linearizes raw pixel-space dynamics, remaining competitive with the non-linear teacher is a strong signal that the finite representation possesses sufficient capacity.
>
> Moreover, we take this opportunity to highlight (as you pointed out before) that the broader value of global linearization goes beyond raw, endpoint FID metrics: our approach's strength is not pure fast sampling; it unlocks **new analytical tools, interpretable spectral decompositions, and fine-grained control** over the teacher CFM model. These capabilities significantly expand the applicability and provide an entirely novel, mathematically principled toolbox for understanding and controlling the dynamics of CFM-based generative models.
>
>  **Inversion Results and the Role of Consistency Loss (W3)**
>
> Thanks for your remark! We understand that evaluating a single inverted static image can be subjective, and that the "without consistency" reconstructions in Fig. 15 might visually appear closer to the original at first glance.
>
> However, as shown in the MSE loss plots below the inversion examples (page 26), the optimization landscape is significantly smoother and more stable with the consistency loss. More importantly, as the "no-consistency" model only learns a shortcut between endpoints, it **does not reliably map to the true generative trajectory**. This is clear visually as noise artifacts appear on the inverted image, and become clearer during downstream tasks: when we port the inverted noise back into the original CFM pipeline for semantic editing, the no-consistency model suffers from noticeable color aberrations and structural artifacts (as demonstrated in Figures 4, 16, and 17).
>
> **Presentation and Typos (W4)**
>
> Thank you for reading the manuscript so carefully. We apologize for the distracting errors. Indeed, Meanflow is not endpoint pair regression, but learns the mapping $x_0 \mapsto x_1$. This shorthand was misleading, and we will correct this. We will correct all typos in the final version.
>
> We will correct all minor presentation issues for the final version.
>
> We will also release our complete implementation to enable follow-up work and ensure reproducibility of all our results.

---

> > ### Author Rebuttal · Reviewer_DZKj · 2026-04-02
> >
> > Thank you to the authors for the clarifications, which have mostly resolved my concerns.
> >
> > One follow-up: why is the 2048+2D FID score (8.9) worse than the 1026D FID in Fig. 5 (~8)?

---

> > > ### Author Response · Authors · 2026-04-06
> > >
> > > Thank you for this follow-up question!
> > >
> > > Indeed, we also observed that 2048+2D FID score (8.9) is slightly worse than the 1026D FID in Fig. 5 (~8).
> > >
> > > Theoretically, with optimally-trained models, the performance can only improve or plateau (e.g., in case the complete dynamics have already fully captured) when increasing the Koopman dimension.
> > >
> > > We attribute the observed slight drop to the fact that we ran the 2048+2D experiment _from scratch_ with all default settings (e.g., learning rate, etc.), which were tuned for the lower dimensions and could be slightly suboptimal for the higher dimension.
> > >
> > > A more robust approach could be to either increase the dimensionality gradually, ensuring the loss only improves, or, alternatively, to perform a proper hyperparameter search for higher dimensional training.
> > >
> > > Interestingly, the slight FID degradation for high Koopman dimensions was also observed in [Berman et al., "One-Step Offline Distillation…" NeurIPS 2025], and our results, with our trajectory-preserving approach, are consistent with theirs.

---

### Official Review · Reviewer_9oGs · 2026-03-13

**Soundness:** 3
**Presentation:** 3
**Significance:** 2
**Originality:** 3
**Overall Recommendation:** 4
**Confidence:** 3

**Summary:**

In this work, the authors aims at linearizing the dynamics of conditional flow matching models by leveraging Koopman Operator theory. In particular, the proposed approach develops a "infinitesimal consistency" loss that aligns the linear dynamics with the teacher’s vector field along the entire path, instead of just endpoints. The author empirically studied the method on MNIST, CIFAR-10, and 32×32 FFHQ with an OT-CFM teacher and the learned one-step sampler generates competitive FIDs and shows superior inference efficiency. The model also offers spectral interpretability through Koopman eigenmodes.

**Compliance With Llm Reviewing Policy:**

Affirmed.

**Final Justification:**

The authors have properly answered my questions, and I have decided to keep the score.

**Key Questions For Authors:**

1. Can the authors elaborate more on how sensitive the estimator is to errors in teacher model's approximation? (Proposition 3)

2. Would the linearization process be significantly more difficult for teacher models with more complex vector fields, such as high-resolution Diffusion Transformers?

3. Are there fundamental challenge in applying the proposed method to higher resolutional image generative like ImageNet?

4. Adding consistency loss improves trajectory MSE but can result in worse FID. Is there a systematic method in selecting $\lambda_{cons}$ to balance the tradeoff?

**Limitations:**

yes

**Strengths And Weaknesses:**

Strengths:

1. The problem is well framed and proposed method with "infinitesimal consistency" loss is novel.

2. The authors include solid ablation study in Koopman space.

3. The proposed method naturally offers nice interpretability property where eigenmodes provide editable semantic meaning.

Weaknesses:

1. Empirical study only includes small scale dataset like MNIST and CIFAR-10. It's hard to tell whether the model works on large scale like ImageNet.

2. Also empirical study applies a pretrained OT-CFM model as the teacher model, which is not the most advanced vision generative models available. This may harm the practical usage of the proposed method.

---

> ### Author Rebuttal · Authors · 2026-03-31
>
> Dear Reviewer 9oGs,
>
> We thank you for your review and for recognizing that our  "'infinitesimal consistency' loss is novel". We are happy that you appreciated the "solid ablation study in Koopman space" and the "nice interpretability property where eigenmodes provide editable semantic meaning" that our method naturally offers through Koopman eigenmodes. Successfully linearizing non-autonomous dynamics required overcoming non-trivial theoretical and practical challenges, and we are glad that you found value in the resulting simulation-free framework.
>
> Below, we address your specific questions and concerns:
>
>
> **Scalability to High-Resolution, ImageNet, and DiTs (W1, Q2, Q3)**
>
> You raised a valid concern regarding the evaluation on smaller-scale datasets and how the method might scale to complex vector fields like Diffusion Transformers (DiTs) or datasets like ImageNet.
>
> - We tested our approach across datasets of increasing complexity: MNIST, CIFAR-10, and FFHQ. Notably, we did not notice any performance degradation from one dataset to another besides requiring a slight network size change (Appendix, Table 3.).
> - To address resolution scalability directly, we point to Appendix C.3, where we demonstrated that our model achieves solid performance relative to the teacher on 64×64×3 FFHQ images (FID of **13.14**).
> - It is important to note that our current pipeline operates on raw pixel space dynamics, which is inherently challenging. Most state-of-the-art models (including high-resolution DiTs) operate in a compressed latent space to handle the high-dimensional complexity.
> - Because our Koopman framework is **data and representation agnostic**, it can operate entirely within these latent representations. In latent space, the dimensionality remains manageable, and linearizing the underlying dynamics is conceptually equivalent to linearizing than raw pixels.
> - Therefore, we see no fundamental theoretical challenge in applying this method to ImageNet or DiTs, a highly promising direction for scaling. We did not pursue it only because of limited time and computational budget, and we will be happy to include such results in the final version.
>
> **Practicality and Choice of Teacher Model (W2)**
>
> Excellent remark! Using a pretrained OT-CFM is an arbitrary choice meant to serve as a proof-of-concept for our methodology. However, our framework is **not restricted** to OT-CFM; it can be applied to other continuous-time dynamics, including various flow-based and diffusion models.
> - For instance, diffusion SDEs can be formulated via the Fokker-Planck equation, which utilizes a velocity field that includes the score function. Our unbiased estimator for the consistency loss can be adapted for this, meaning there are no fundamental theoretical limits to applying our method to more advanced generative teachers.
> - Otherwise, any flow matching model can be directly plugged into our framework.
>
> **Estimator Sensitivity to Teacher Errors (Q1)**
>
> Regarding the sensitivity of the estimator in Proposition 3, the estimator itself, as we prove, is **mathematically unbiased**.
>
> - Consequently, any approximation errors inherently present in the teacher model's velocity field will naturally be transferred to the learned Koopman operator and observables.
> - We view this as a limitation of the specific teacher model's fidelity rather than a limitation of our linearizing Koopman pipeline. Moreover, we also show that we can quantify the difference between two teachers: we apply this idea in Section 6.4, Insights on teacher training: we observe how Koopman modes of the final CFM are added during training of the CFM.
>
> **Balancing the Trajectory MSE vs. FID Tradeoff (Q4)**
>
> You correctly noted that while the consistency loss improves trajectory MSE, it can impact the final FID. We emphasize that this behavior is expected, as we linearize the **full trajectory** — something that FID, which only evaluates the final output, cannot capture. Thus, one-step distillation naturally produces better FID, but does not enable the types of applications that our framework provides.
>
> We balance loss terms to maintain comparable magnitudes during training. This heuristic yields stable optimization and strong empirical performance, including on downstream applications. We did not conduct an exhaustive search over weighting schemes due to time and budget constraints. We will include an ablation of loss weights in the final version.

---

> > ### Author Rebuttal · Reviewer_9oGs · 2026-04-02
> >
> > I thank the authors for the responses which have answered my main questions. I stay positive about this work.

---

### Official Review · Reviewer_yQSA · 2026-03-16

**Soundness:** 3
**Presentation:** 2
**Significance:** 1
**Originality:** 2
**Overall Recommendation:** 3
**Confidence:** 3

**Summary:**

The authors present an operator framework based on Koopman operator theorey that proposes a linearization of Conditional Flow Matching. The motivation is to achieve onestep image generative modeling that is also more interpretable. They carry out experiments investicating if the sampler achieves trajectory fidelity while keeping sampling performance, if their consistency loss is crucial for learning an interpretable linearization and if the learned Koopman latent space leads to a more robust and functionally useful model.

**Compliance With Llm Reviewing Policy:**

Affirmed.

**Final Justification:**

I find the work of good quality, nevertheless it is my estimate that the limitations in significance and some major improvments required in the scientific delivery of this work require one more proper iteration before the work is to be published in this venue.

**Key Questions For Authors:**

1. What are the main limitations of the method in regards to its scalability, can you also discuss how and if the method can be used in the broad generative modeling applications?

2. what is the significance of?  "low-energy modes are largely shared across classes, while higher-energy modes differentiate them"? Can also try to discuss your analysis a bit further?

3. How do you access that the model is "functionally useful" in this context?

**Limitations:**

There is a very short limitations discussion, this work needs a proper limitations discussion. This is crucial for these type of works.

**Strengths And Weaknesses:**

Strengths:
1. The paper gives a good motivation for the proposed learning of a global Koopman linearization of the non-autonomous dynamics in Conditional Flow Matching models. The authors bring sound arguments to its implementation as well.

2. The previous works are well presented as well as the method is clearly structured and explained, especially the Koopman Theory for autonomous systems section.

3. The work provides some insights and increases the understanding of the significance of the latent space of CFMs.

Weaknesses:

1. The presentation quality of the last sections is not so consistent. For example the interpretability analysis section looks a bit rushed making the analysis tedius to follow as there is not enough discussion to elaborate on different aspectes of the analysis. Also some figure captions are too short. As a minor point some of the plot fonts are too small to read

2. From my perspective the singificance of this work is somewhat limited. Linearization of the latent space trajectories is interesting but it has been broadly explored and its impact is up to now limited as well.

3. The experimentation is a bit limited to fully appreciate that semandic content is captured.

---

> ### Author Rebuttal · Authors · 2026-03-31
>
> Dear Reviewer yQSA,
>
> We thank you for your thoughtful review and for recognizing the "good motivation" of the method,  "clearly structured and explained", where "insights ... increase the understanding of the significance of the latent space of CFMs." We highlight that establishing an unbiased, simulation-free estimator (Proposition 3) is a non-trivial theoretical result that enables this framework to inherit the efficient training strategy of CFM.
>
> ## Weaknesses:
>
> **W1: Presentation.**
> Thank you for this remark! Indeed, balancing the theoretical foundations with the extensive practical explorations within the page limit presented a significant challenge, pushing key results to the appendix or compressing them. We will restructure Section 5 and 6, expand the figure captions to make them self-contained, increase the font sizes on all mentioned plots, and integrate critical qualitative results from the appendix into the main text.
>
> **W2: Significance of the work.**
> While latent space linearization has been explored in other contexts, we are the first to use **continuous** Koopman theory and propose a theoretically-founded approach that **fully linearizes** trajectories.  The significance of our work lies in bridging this theoretical gap to unlock highly practical, plug-and-play tools for the ML community, while significantly extending recent work [N. Berman et al. NeurIPS 2025]. Specifically, our novel global linearization leads to:
>
> - **Semantic coherent latent space (section 6.4):** We show quantitatively and qualitatively that it enables identifying semantic directions in latent space, which fails when using endpoint distillation (no consistency).
> - **Analyze Teacher Dynamics (Figure 3, section 6.4 - Class-conditioned signatures):** We show that the model generates coarse-to-fine details using Koopman modes. We also show that modes are used differently across classes.
> - **Training insights (section 6.4):**  Across teacher checkpoints, we observe a spectral bias during training, where teacher learns Koopman modes ordered by their real part Re(λ).
> - **Bridge Latent and Pixel Space (Noise Engineering):** Because our method truly linearizes the trajectory (unlike endpoint distillation), there is a commutative structure between the CFM space and the Koopman space. Insights gained in the Koopman space can be directly applied to the original CFM.
>
> We believe that more exploration in this direction can stem from our work.
>
> **W3: Limited experimentation**
> We understand that evaluating semantic disentanglement can be subjective. In Table 7, we provide rigorous CLIP and LPIPS scores across multiple attributes (Hat, Sunglasses, Smile, Age, Gender) comparing models with and without our consistency loss. The consistency-trained Koopman modes align with semantic directions at a significantly higher rate. We will make these observations clearer.
>
> ## Questions:
>
> **Q1**
> We will add a detailed Limitations section. The primary limitations are:
>
> - **Data Requirements:** We currently use pre-generated start-and-end pairs. While our unbiased estimator makes intermediate trajectory sampling simulation-free, avoiding the pair generation entirely is an open challenge, potentially solvable via novel operator regularization.
> - **Scaling to ImageNet/High-Res:** Applying our approach to high-resolution images is computationally heavy, as for other generative approaches. Modern generative models operate within a compressed latent space. Deploying our framework on top of these latent spaces presents no fundamental obstacles and is a highly promising direction for scaling, which we did not pursue only because of limited time and computational budget.
>
> **Q2**
> Global linearization enables standard signal processing of the generative process. We propose to view the dataset classes as "filters" applied to the Koopman spectrum (Eq. 16 and 17).
>
> **Significance:** We found that low-energy modes are not amplified or decayed by specific classes; they are _shared_ across the dataset, encoding global structures (like background or basic shapes). Conversely, high-energy modes act as class-specific discriminators. For instance, our transfer functions reveal that visually similar classes (like 'plane' and 'bird') amplify similar high-energy modes. This provides a novel, data-driven way to analyze class similarities in the generative process.
>
> **Q3**
> We assess functional usefulness through the model's performance on standard downstream vision tasks. As detailed in Appendix F.4 (which we will highlight more prominently in the main text), our consistency-trained model significantly outperforms the purely distilled baseline on:
>
> - **Inpainting, Super-Resolution, Denoising:** The structured, semantic decomposition of our Koopman modes allows the model to successfully reconstruct corrupted images.
> - **Inversion, Editing:** We achieve latent inversion with noticeably fewer artifacts and smoother optimization landscape compared to standard flow inversion.

---

> > ### Author Rebuttal · Reviewer_yQSA · 2026-04-03
> >
> > Thank you for the clarifications, you have addressed most of my concerns in terms of the analysis and significance.  I understand now the Semantic coherent latent space, thank you for explaining. My estimate is that the major limitation is the significance of the method and a secondary concern is the presentation as adequate and precise explanations are an integral part of a proposed conference paper the gap maybe too large to be filled within the scope of a rebuttal.

---

> > > ### Author Response · Authors · 2026-04-06
> > >
> > > Thank you for your acknowledgement. We appreciate the reviewer's comments. While we do believe that our work presents a principled and technically novel interpretability and analysis approach for multistep generative models (as also acknowledged by the other reviewers), we nevertheless respect your position.
> > >
> > > In regards to the presentation, as mentioned in our rebuttal and other responses, we fully commit to including the additional experiments that we conducted and incorporating all presentation-related changes and suggestions (such as addressing all typos and improving the notational consistency) for the possible final version.
> > >
> > > Additionally, we will release our complete implementation to enable _full_ reproducibility of all of our results and to enable follow-up work.
> > >
> > > Thank you!

---

### Decision · Program_Chairs · 2026-04-30

**Decision:**

Accept (regular)

**Comment:**

This paper studies continuous normalizing flows with Koopman operator theory. In particular, the authors train a non-linear encoder that maps states along the flow path to their corresponding Koopman coordinates, and in this Koopman space the flow dynamics is linear and can therefore be described via a simple matrix and solved efficiently. This linearization has additional benefits for interpretability and controllability, as the authors show.

The reviewers pointed out the conceptual novelty of the approach and considered the approach to be well-motivated. In particular the interpretability results were highlighted. Nonetheless, some concerns were raised, in particular with respect to the scalability of the method. The authors apply their approach only on relative simple image generation tasks. Furthermore, some reviewers pointed out that the presentation could be improved. Further questions could be answered during the rebuttal process.

While I agree that the paper would benefit from more large-scale experiments, I believe that the method in its current form is novel and interesting, and the authors promised to update the presentation as described during the rebuttal. Efficient generation is an important problem, which the paper tackles from a new angle through linearization via Koopman operators, and interpretability and controllability is equally relevant. For these reasons, I believe the paper is interesting to the ICML community and might inspire follow-up work. Hence, I suggest the paper for acceptance.